# Generative Modeling through the Semi-dual Formulation of Unbalanced Optimal Transport

**Jaemoo Choi**[*]
Seoul National University
toony42@snu.ac.kr

**Jaewoong Choi** [*]
Korea Institute for Advanced Study
chwj1475@kias.re.kr

**Myungjoo Kang**
Seoul National University
mkang@snu.ac.kr

## Abstract

Optimal Transport (OT) problem investigates a transport map that bridges two distributions while minimizing a given cost function. In this regard, OT between tractable prior distribution and data has been utilized for generative modeling tasks. However, OT-based methods are susceptible to outliers and face optimization challenges during training. In this paper, we propose a novel generative model based on the semi-dual formulation of Unbalanced Optimal Transport (UOT). Unlike OT, UOT relaxes the hard constraint on distribution matching. This approach provides better robustness against outliers, stability during training, and faster convergence. We validate these properties empirically through experiments. Moreover, we study the theoretical upper-bound of divergence between distributions in UOT. Our model outperforms existing OT-based generative models, achieving FID scores of 2.97 on CIFAR-10 and 6.36 on CelebA-HQ-256. The code is available at https://github.com/Jae-Moo/UOTM.

## 1 Introduction

Optimal Transport theory [57, 77] explores the cost-optimal transport to transform one probability distribution into another. Since WGAN [6], OT theory has attracted significant attention in the field of generative modeling as a framework for addressing important challenges in this field. In particular, WGAN introduced the Wasserstein distance, an OT-based probability distance, as a loss function for optimizing generative models. WGAN measures the Wasserstein distance between the data distribution and the generated distribution, and minimizes this distance during training. The introduction of OT-based distance has improved the diversity [6, 27], convergence [63], and stability [36, 52] of generative models, such as WGAN [6] and its variants [27, 45, 56]. However, several works showed that minimizing the Wasserstein distance still faces computational challenges, and the models tend to diverge without a strong regularization term, such as gradient penalty [51, 63].

Recently, there has been a surge of research on directly modeling the optimal transport map between the input prior distribution and the real data distribution [2, 3, 49, 61, 82]. In other words, the optimal transport serves as the generative model itself. These approaches showed promising results that are comparable to WGAN models. However, the classical optimal transport-based approaches are known to be highly sensitive to outlier-like samples [8]. The existence of a few outliers can have a significant impact on the overall OT-based distance and the corresponding transport map. This sensitivity can be problematic when dealing with large-scale real-world datasets where outliers and noises are inevitable.

To overcome these challenges, we suggest a new generative algorithm utilizing the semi-dual formulation of the Unbalanced Optimal Transport (UOT) problem [12, 44]. In this regard, we refer to our model as the *UOT-based generative model (UOTM)*. The UOT framework relaxes the hard constraints

---

[*]Equal contribution. Correspondence to: Myungjoo Kang <mkang@snu.ac.kr>.

37th Conference on Neural Information Processing Systems (NeurIPS 2023).

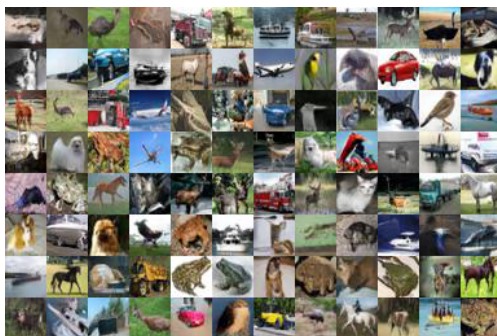 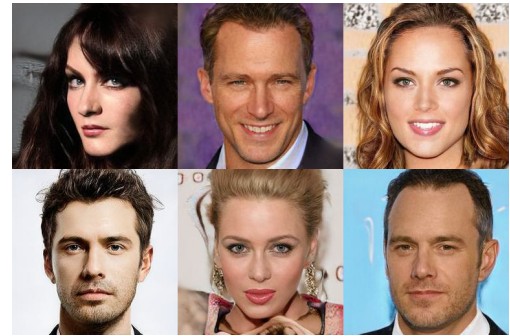

Figure 1: **Generated Samples from UOTM** trained on **Left**: CIFAR-10 and **Right**: CelebA-HQ.

of marginal distribution in the OT framework by introducing soft entropic penalties. This soft constraint provides additional robustness against outliers [8, 66]. Our experimental results demonstrate that UOTM exhibits such outlier robustness, as well as faster and more stable convergence compared to existing OT-based models. Particularly, this better convergence property leads to a tighter matching of data distribution than the OT-based framework despite the soft constraint of UOT. Our UOTM achieves FID scores of 2.97 on CIFAR-10 and 6.36 on CelebA-HQ, outperforming existing OT-based adversarial methods by a significant margin and approaching state-of-the-art performance. Furthermore, this decent performance is maintained across various objective designs. Our contributions can be summarized as follows:

- UOTM is the first generative model that utilizes the semi-dual form of UOT.
- We analyze the theoretical upper-bound of divergence between marginal distributions in UOT, and validate our findings through empirical experiments.
- We demonstrate that UOTM presents outlier robustness and fast and stable convergence.
- To the best of our knowledge, UOTM is the first OT-based generative model that achieves near state-of-the-art performance on real-world image datasets.

## 2 Background

**Notations**   Let $\mathcal{X}$, $\mathcal{Y}$ be two compact complete metric spaces, $\mu$ and $\nu$ be probability distributions on $\mathcal{X}$ and $\mathcal{Y}$, respectively. We regard $\mu$ and $\nu$ as the source and target distributions. In generative modeling tasks, $\mu$ and $\nu$ correspond to *tractable noise* and *data distributions*. For a measurable map $T$, $T_{\#}\mu$ represents the pushforward distribution of $\mu$. $\Pi(\mu, \nu)$ denote the set of joint probability distributions on $\mathcal{X} \times \mathcal{Y}$ whose marginals are $\mu$ and $\nu$, respectively. $\mathcal{M}_{+}(\mathcal{X} \times \mathcal{Y})$ denote the set of joint positive measures defined on $\mathcal{X} \times \mathcal{Y}$. For convenience, for $\pi \in \mathcal{M}_{+}(\mathcal{X} \times \mathcal{Y})$, let $\pi_0(x)$ and $\pi_1(y)$ be the marginals with respect to $\mathcal{X}$ and $\mathcal{Y}$. $c(x, y)$ refers to the transport cost function defined on $\mathcal{X} \times \mathcal{Y}$. Throughout this paper, we consider $\mathcal{X} = \mathcal{Y} \subset \mathbb{R}^d$ with the quadratic cost, $c(x, y) = \tau \|x - y\|_2^2$, where $d$ indicates the dimension of data. Here, $\tau$ is a given positive constant. For the precise notations and assumptions, see Appendix A.

**Optimal Transport (OT)**   OT addresses the problem of searching for the most cost-minimizing way to transport source distribution $\mu$ to target $\nu$, based on a given cost function $c(\cdot, \cdot)$. In the beginning, Monge [53] formulated this problem with a deterministic transport map. However, this formulation is non-convex and can be ill-posed depending on the choices of $\mu$ and $\nu$. To alleviate these problems, the Kantorovich OT problem [33] was introduced, which is a convex relaxation of the Monge problem. Formally, the OT cost of the relaxed problem is given by as follows:

$$C(\mu, \nu) := \inf_{\pi \in \Pi(\mu, \nu)} \left[ \int_{\mathcal{X} \times \mathcal{Y}} c(x, y) d\pi(x, y) \right], \tag{1}$$

where $c$ is a cost function, and $\pi$ is a coupling of $\mu$ and $\nu$. Unlike Monge problem, the minimizer $\pi^{\star}$ of Eq 1 always exists under some mild assumptions on $(\mathcal{X}, \mu)$, $(\mathcal{Y}, \nu)$ and the cost $c$ ([77], Chapter 5). Then, the ***dual form*** of Eq 1 is given as:

$$C(\mu, \nu) = \sup_{u(x)+v(y) \leq c(x,y)} \left[ \int_{\mathcal{X}} u(x) d\mu(x) + \int_{\mathcal{Y}} v(y) d\nu(y) \right], \tag{2}$$

where $u$ and $v$ are Lebesgue integrable with respect to measure $\mu$ and $\nu$, i.e., $u \in L^1(\mu)$ and $v \in L^1(\nu)$. For a particular case where $c(x, y)$ is equal to the distance function between $x$ and $y$, then $u = -v$ and $u$ is 1-Lipschitz [77]. In such case, we call Eq 2 a *Kantorovich-Rubinstein duality*. For the general cost $c(\cdot, \cdot)$, the Eq 2 can be reformulated as follows ([77], Chapter 5):

$$C(\mu, \nu) = \sup_{v \in L^1(\nu)} \left[ \int_{\mathcal{X}} v^c(x) d\mu(x) + \int_{\mathcal{Y}} v(y) d\nu(y) \right],  \qquad (3)$$

where the $c$-transform of $v$ is defined as $v^c(x) = \inf_{y \in \mathcal{Y}} (c(x, y) - v(y))$. We call this formulation (Eq 3) a *semi-dual formulation of OT*.

**Unbalanced Optimal Transport (UOT)**  Recently, a new type of optimal transport problem has emerged, which is called *Unbalanced Optimal Transport (UOT)* [12, 44]. The *Csiszàr divergence* $D_f(\mu|\nu)$ associated with $f$ is a generalization of $f$-divergence for the case where $\mu$ is not absolutely continuous with respect to $\nu$ (See Appendix A for the precise definition). Here, the entropy function $f : [0, \infty) \to [0, \infty]$ is assumed to be convex, lower semi-continuous, and non-negative. Note that the $f$-divergence families include a wide variety of divergences, such as Kullback-Leibler (KL) divergence and $\chi^2$ divergence. (See [71] for more examples of $f$-divergence and its corresponding generator $f$.) Formally, the UOT problem is formulated as follows:

$$C_{ub}(\mu, \nu) := \inf_{\pi \in \mathcal{M}_+(\mathcal{X} \times \mathcal{Y})} \left[ \int_{\mathcal{X} \times \mathcal{Y}} c(x, y) d\pi(x, y) + D_{\Psi_1}(\pi_0|\mu) + D_{\Psi_2}(\pi_1|\nu) \right].  \qquad (4)$$

The UOT formulation (Eq 4) has two key properties. First, UOT can handle the transportation of any positive measures by relaxing the marginal constraint [12, 23, 44, 57], allowing for greater flexibility in the transport problem. Second, UOT can address the sensitivity to outliers, which is a major limitation of OT. In standard OT, the marginal constraints require that even outlier samples are transported to the target distribution. This makes the OT objective (Eq 1) to be significantly affected by a few outliers. This sensitivity of OT to outliers implies that the OT distance between two distributions can be dominated by these outliers [8, 66]. On the other hand, UOT can exclude outliers from the consideration by flexibly shifting the marginals. Both properties are crucial characteristics of UOT. However, in this work, we investigate a generative model that transports probability measures. Because the total mass of probability measure is equal to 1, we focus on the latter property.

## 3 Method

### 3.1 Dual and Semi-dual Formulation of UOT

Similar to OT, UOT also provides a *dual formulation* [12, 21, 72]:

$$C_{ub}(\mu, \nu) = \sup_{u(x)+v(y) \leq c(x,y)} \left[ \int_{\mathcal{X}} -\Psi_1^*(-u(x)) d\mu(x) + \int_{\mathcal{Y}} -\Psi_2^*(-v(y)) d\nu(y) \right],  \qquad (5)$$

with $u \in \mathcal{C}(\mathcal{X})$, $v \in \mathcal{C}(\mathcal{Y})$ where $\mathcal{C}$ denotes a set of continuous functions over its domain. Here, $f^*$ denotes the *convex conjugate* of $f$, i.e., $f^*(y) = \sup_{x \in \mathbb{R}} \{\langle x, y \rangle - f(x)\}$ for $f : \mathbb{R} \to [-\infty, \infty]$.

**Remark 3.1** (**UOT as a Generalization of OT**). Suppose $\Psi_1$ and $\Psi_2$ are the convex indicator function of $\{1\}$, i.e. have zero values for $x = 1$ and otherwise $\infty$. Then, the objective in Eq 4 becomes infinity if $\pi_0 \neq \mu$ or $\pi_1 \neq \nu$, which means that UOT reduces into classical OT. Moreover, $\Psi_1^*(x) = \Psi_2^*(x) = x$. If we replace $\Psi_1^*$ and $\Psi_2^*$ with the identity function, Eq 5 precisely recovers the dual form of OT (Eq 2). Note that the *hard constraint corresponds to $\Psi^*(x) = x$*.

Now, we introduce the *semi-dual formulation of UOT* [72]. For this formulation, we assume that $\Psi_1^*$ and $\Psi_2^*$ are non-decreasing and differentiable functions ($f^*$ is non-decreasing for the non-negative entropy function $f$ [65]):

$$C_{ub}(\mu, \nu) = \sup_{v \in \mathcal{C}} \left[ \int_{\mathcal{X}} -\Psi_1^*(-v^c(x))) d\mu(x) + \int_{\mathcal{Y}} -\Psi_2^*(-v(y)) d\nu(y) \right],  \qquad (6)$$

where $v^c(x)$ denotes the $c$-transform of $v$ as in Eq 3.

**Remark 3.2** ($\Psi^*$ **Candidate**). Here, we clarify the feasible set of $\Psi^*$ in the semi-dual form of UOT. As aforementioned, the assumption for deriving Eq 6 is that $\Psi^*$ is a non-decreasing, differentiable function. Recall that for any function $f$, its convex conjugate $f^*$ is convex and lower semi-continuous [9]. Also, for any convex and lower semi-continuous $f$, $f$ is a convex conjugate of $f^*$ [7]. Combining these two results, *any non-decreasing, convex, and differentiable function can be a candidate of $\Psi^*$.*

## 3.2 Generative Modeling with the Semi-dual Form of UOT

In this section, we describe how we implement a generative model based on the semi-dual form (Eq 6) of UOT. Following [18, 28, 41, 54, 61], we introduce $T_v$ to approximate $v^c$ as follows:

$$T_v(x) \in \operatorname*{arginf}_{y \in \mathcal{Y}} \left[ c(x, y) - v(y) \right] \quad \Leftrightarrow \quad v^c(x) = c\left(x, T_v(x)\right) - v\left(T_v(x)\right), \tag{7}$$

Note that $T_v$ is measurable ([10], Prop 7.33). Then, the following objective $J(v)$ can be derived from the equation inside supremum in Eq (6) and the right-hand side of Eq 7:

$$J(v) := \int_{\mathcal{X}} \Psi_1^* \left( - \left[ (c(x, T_v(x)) - v\left(T_v(x)\right)) \right] \right) d\mu(x) + \int_{\mathcal{Y}} \Psi_2^* \left(-v(y)\right) d\nu(y). \tag{8}$$

In practice, there is no closed-form expression of the optimal $T_v$ for each $v$. Hence, the optimization $T_v$ for each $v$ is required as in the generator-discriminator training of GAN [25]. In this work, we parametrize $v = v_\phi$ and $T_v = T_\theta$ with neural networks with parameter $\phi$ and $\theta$. Then, our learning objective $\mathcal{L}_{v_\phi, T_\theta}$ can be formulated as follows:

$$\mathcal{L}_{v_\phi, T_\theta} = \inf_{v_\phi} \left[ \int_{\mathcal{X}} \Psi_1^* \left( - \inf_{T_\theta} \left[ c\left(x, T_\theta(x)\right) - v_\phi\left(T_\theta(x)\right) \right] \right) d\mu(x) + \int_{\mathcal{Y}} \Psi_2^* \left(-v_\phi(y)\right) d\nu(y) \right]. \tag{9}$$

Finally, based on Eq 9, we propose a new training algorithm (Algorithm 1), called the *UOT-based generative model (UOTM)*. Similar to the training procedure of GANs, we alternately update the potential $v_\phi$ (lines 2-4) and generator $T_\theta$ (lines 5-7).

---

**Algorithm 1** Training algorithm of UOTM

---

**Require:** The source distribution $\mu$ and the target distribution $\nu$. Non-decreasing, differentiable, convex function pair $(\Psi_1^*, \Psi_2^*)$. Generator network $T_\theta$ and the discriminator network $v_\phi$. Total iteration number $K$.
1: **for** $k = 0, 1, 2, \ldots, K$ **do**
2:      Sample a batch $X \sim \mu$, $Y \sim \nu$, $z \sim \mathcal{N}(\mathbf{0}, \mathbf{I})$.
3:      $\mathcal{L}_v = \frac{1}{|X|} \sum_{x \in X} \Psi_1^* \left(-c\left(x, T_\theta(x, z)\right) + v_\phi\left(T_\theta(x, z)\right)\right) + \frac{1}{|Y|} \sum_{y \in Y} \Psi_2^*(-v_\phi(y))$
4:      Update $\phi$ by using the loss $\mathcal{L}_v$.
5:      Sample a batch $X \sim \mu$, $z \sim \mathcal{N}(\mathbf{0}, \mathbf{I})$.
6:      $\mathcal{L}_T = \frac{1}{|X|} \sum_{x \in X} \left(c\left(x, T_\theta(x, z)\right) - v_\phi(T_\theta(x, z))\right)$.
7:      Update $\theta$ by using the loss $\mathcal{L}_T$.
8: **end for**

---

Following [26, 42], we add stochasticity in our generator $T_\theta$ by putting an auxiliary variable $z$ as an additional input. This allows us to obtain the stochastic transport plan $\pi(\cdot|x)$ for given input $x \in \mathcal{X}$, motivated by the Kantorovich relaxation [33]. The role of the auxiliary variable has been extensively discussed in the literature [13, 26, 42, 80], and it has been shown to be useful in generative modeling. Moreover, we incorporate $R_1$ regularization [60], $\mathcal{L}_{reg} = \lambda \|\nabla_x v_\phi(y)\|_2^2$ for real data $y \in \mathcal{Y}$, into the objective (Eq 9), which is a popular regularization method employed in various studies [13, 51, 80].

## 3.3 Some Properties of UOTM

**Divergence Upper-Bound of Marginals**     UOT relaxes the hard constraint of marginal distributions in OT into the soft constraint with the Csiszàr divergence regularizer. Therefore, a natural question is *how much divergence is incurred in the marginal distributions* because of this soft constraint in UOT. The following theorem proves that the upper-bound of divergences between the marginals in UOT is linearly proportional to $\tau$ in the cost function $c(x, y)$. This result follows our intuition because down-scaling $\tau$ is equivalent to the relative up-scaling of divergences $D_{\Psi_1}$, $D_{\Psi_2}$ in Eq 4. (Notably, in our experiments, the optimization benefit outweighed the effect of soft constraint (Sec 5.2).)

**Theorem 3.3.** *Suppose that $\mu$ and $\nu$ are probability densities defined on $\mathcal{X}$ and $\mathcal{Y}$. Given the assumptions in Appendix A, suppose that $\mu, \nu$ are absolutely continuous with respect to Lebesgue measure and $\Psi^*$ is continuously differentiable. Assuming that the optimal potential $v^\star = \inf_{v \in \mathcal{C}} J(v)$ exists, $v^\star$ is a solution of the following objective*

$$\tilde{J}(v) = \int_{\mathcal{X}} -v^c(x) d\tilde{\mu}(x) + \int_{\mathcal{Y}} -v(y) d\tilde{\nu}(y), \tag{10}$$

*where $\tilde{\mu}(x) = \Psi_1^{*\prime}(-v^{\star c}(x))\mu(x)$ and $\tilde{\nu}(y) = \Psi_2^{*\prime}(-v^\star(y))\nu(y)$. Note that the assumptions guarantee the existence of optimal transport map $T^\star$ between $\tilde{\mu}$ and $\tilde{\nu}$. Furthermore, $T^\star$ satisfies*

$$T^\star(x) \in \arginf_{y \in \mathcal{Y}} [c(x,y) - v^\star(y)], \tag{11}$$

*$\mu$-almost surely. In particular, $D_{\Psi_1}(\tilde{\mu}|\mu) + D_{\Psi_2}(\tilde{\nu}|\nu) \leq \tau \mathcal{W}_2^2(\mu, \nu)$ where $\mathcal{W}_2(\mu, \nu)$ is a Wasserstein-2 distance between $\mu$ and $\nu$.*

Theorem 3.3 shows that $T_{v^\star}$ (Eq 7) is a valid parametrization of the optimal transport map $T^\star$ that transports $\tilde{\mu}$ to $\tilde{\nu}$. Moreover, if $\tau$ is sufficiently small, then $\tilde{\mu}$ and $\tilde{\nu}$ are close to $\mu$ and $\nu$, respectively. Therefore, we can infer that $T_{v^\star}$ will transport $\mu$ to a distribution that is similar to $\nu$.

**Stable Convergence**    We discuss convergence properties of UOT objective $J$ and the corresponding OT objective $\tilde{J}$ for the potential $v$ through the theoretical findings of Gallouët et al. [21].

**Theorem 3.4** ([21]). *Under some mild assumptions in Appendix A, the following holds:*

$$J(v) - J(v^\star) \geq \frac{1}{2\lambda} \mathbb{E}_{\tilde{\mu}} \left[ \|\nabla (v^c - v^{\star c})\|_2^2 \right] + C_1 \mathbb{E}_\mu \left[ (v^c - v^{\star c})^2 \right] + C_2 \mathbb{E}_\nu \left[ (v - v^\star)^2 \right], \tag{12}$$

*for some positive constant $C_1$ and $C_2$. Furthermore, $\tilde{J}(v) - \tilde{J}(v^\star) \geq \frac{1}{2\lambda} \mathbb{E}_{\tilde{\mu}} \left[ \|\nabla (v^c - v^{\star c})\|_2^2 \right]$.*

Theorem 3.4 suggests that the UOT objective $J$ gains stability over the OT objective $\tilde{J}$ in two aspects. First, while $\tilde{J}$ only bounds the gradient error $\|\nabla (v^c - v^{\star c})\|_2^2$, $J$ also bounds the function error $(v^c - v^{\star c})^2$. Second, $J$ provides control over both the original function $v$ and its $c$-transform $v^c$ in $L^2$ sense. We hypothesize this stable convergence property conveys practical benefits in neural network optimization. In particular, UOTM attains better distribution matching despite its soft constraint (Sec 5.2) and faster convergence during training (Sec 5.3).

## 4    Related Work

**Optimal Transport**    Optimal Transport (OT) problem addresses a transport map between two distributions that minimizes a specified cost function. This OT map has been extensively utilized in various applications, such as generative modeling [2, 45, 61], point cloud approximation [50], and domain adaptation [20]. The significant interest in OT literature has resulted in the development of diverse algorithms based on different formulations of OT problem, e.g., primary (Eq 1), dual (Eq 2), and semi-dual forms (Eq 3). First, several works were proposed based on the primary form [47, 81]. These approaches typically involved multiple adversarial regularizers, resulting in a complex and challenging training process. Hence, these methods often exhibited sensitivity to hyperparameters. Second, the relationship between the OT map and the gradient of dual potential led to various dual form based methods. In specific, when the cost function is quadratic, the OT map can be represented as the gradient of the dual potential [77]. This correspondence motivated a new methodology that parameterized the dual potentials to recover the OT map [2, 64]. Seguy et al. [64] introduced the entropic regularization to obtain the optimal dual potentials $u$ and $v$, and extracted the OT map from them via the barycentric projection. Some methods [40, 49] explicitly utilized the convexity of potential by employing input-convex neural networks [1].

Recently, Korotin et al. [39] demonstrated that the semi-dual approaches [18, 61] are the ones that best approximate the OT map among existing methods. These approaches [18, 61] properly recovered OT maps and provided high performance in image generation and image translation tasks for large-scale datasets. Because our UOTM is also based on the semi-dual form, these methods show some connections to our work. If we let $\Psi_1^*$ and $\Psi_2^*$ in Eq 9 as identity functions, our Algorithm 1 reduces to the training procedure of Fan et al. [18] (See Remark 3.2). For convenience, we denote Fan et al. [18] as a *OT-based generative model (OTM)*. Moreover, note that Rout et al. [61] can be considered as a minor modification of parametrization from OTM. In this paper, we denote Rout et al. [61] as *Optimal Transport Modeling (OTM\*)*, and regard it as one of the main counterparts for comparison.

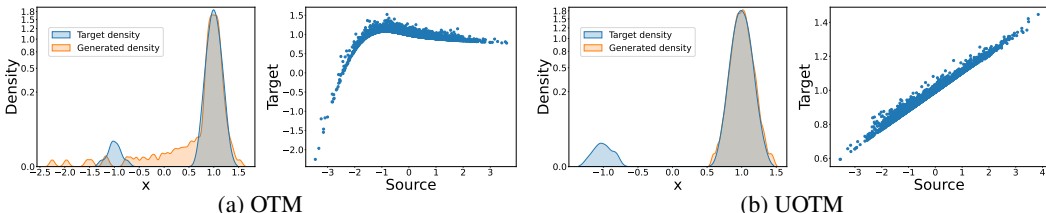

Figure 2: **Outlier Robustness Test on Toy dataset** with 1% outlier. For each subfigure, **Left**: Comparison of target density $\nu$ and generated density $T_{\#}\mu$ and **Right**: Transport map of the trained model $(x, T(x))$. While attempting to fit the outlier distribution, OTM generates undesired samples outside $\nu$ and learns the non-optimal transport map. In contrast, UTOM mainly generates in-distribution samples and achieves the optimal transport map. (For better visualization, the y-scale of the density plot is manually adjusted.)

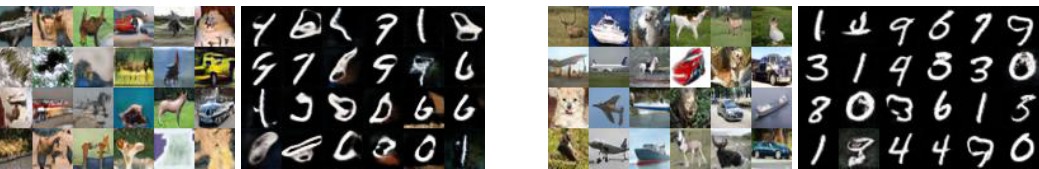

Figure 3: **Outlier Robustness Test on Image dataset** (CIFAR-10 + 1% MNIST). **Left**: OTM exhibits artifacts on both in-distribution and outlier samples. **Right**: UOTM attains higher-fidelity samples while generating MNIST-like samples more sparingly, around 0.2%. FID scores of CIFAR-10-like samples are 13.82 for OTM and 4.56 for UOTM, proving that UOTM is more robust to outliers.

**Unbalanced Optimal Transport** The primal problem of UOT (Eq 4) relaxes hard marginal constraints of Kantorovich's problem (Eq 1) through soft entropic penalties [12]. Most of the recent UOT approaches [11, 19, 48, 58] estimate UOT potentials on discrete space by using the dual formulation of the problem. For example, Pham et al. [58] extended the Sinkhorn method to UOT, and Lübeck et al. [48] learned dual potentials through cyclic properties. To generalize UOT into the continuous case, Yang and Uhler [82] suggested the GAN framework for the primal formulation of unbalanced Monge OT. This method employed three neural networks, one for a transport map, another for a discriminator, and the third for a scaling factor. Balaji et al. [8] claimed that implementing a GAN-like procedure using the dual form of UOT is hard to optimize and unstable. Instead, they proposed an alternative dual form of UOT, which resembles the dual form of OT with a Lagrangian regularizer. Nevertheless, Balaji et al. [8] still requires three different neural networks as in [82]. To the best of our knowledge, our work is the first generative model that leverages the semi-dual formulation of UOT. This formulation allows us to develop a simple but novel adversarial training procedure that does not require the challenging optimization of three neural networks.

## 5 Experiments

In this section, we evaluate our model on the various datasets to answer the following questions:

1. Does UOTM offer more robustness to outliers than OT-based model (OTM)? (§5.1)
2. Does the UOT map accurately match the source distribution to the target distribution? (§5.2)
3. Does UOTM provide stable and fast convergence? (§5.3)
4. Does UOTM provide decent performance across various choices of $\Psi_1^*$ and $\Psi_2^*$? (§5.4)
5. Does UOTM show reasonable performance even without a regularization term $\mathcal{L}_{reg}$? (§5.4)
6. Does UOTM provide decent performance across various choices of $\tau$ in $c(x, y)$? (§5.4)

Unless otherwise stated, we set $\Psi := \Psi_1 = \Psi_2$ and $D_\Psi$ as a *KL divergence* in our UOTM model (Eq 9). The source distribution $\mu$ is a standard Gaussian distribution $\mathcal{N}(\mathbf{0}, \mathbf{I})$ with the same dimension as the target distribution $\nu$. In this case, the entropy function $\Psi$ is given as follows:

$$\Psi(x) = \begin{cases} x \log x - x + 1, & \text{if } x > 0 \\ \infty, & \text{if } x \leq 0 \end{cases}, \qquad \Psi^*(x) = e^x - 1. \tag{13}$$

In addition, when the generator $T_\theta$ and potential $v_\phi$ are parameterized by the same neural networks as OTM* [61], we call them a *small* model. When we use the same network architecture in RGM

Table 1: **Target Distribution Matching Test.** UOTM achieves a better approximation of target distribution $\nu$, i.e., $T_\#\mu \approx \nu$. † indicates the results conducted by ourselves.

| Model | Toy ($\tau = 0.1$) | Toy ($\tau = 0.02$) | CIFAR-10 |
|---|---|---|---|
| Metric | $D_{KL}(T_\#\mu|\nu)$ ($\downarrow$) | | FID ($\downarrow$) |
| OTM† | 0.05 | 0.05 | 7.68 |
| Fixed-$\mu$† | **0.02** | **0.004** | 7.53 |
| **UOTM**† | **0.02** | 0.005 | **2.97** |

Table 3: **Image Generation on CelebA-HQ.**

| Class | Model | FID ($\downarrow$) |
|---|---|---|
| **Diffusion** | Score SDE (VP) [69] | 7.23 |
| | Probability Flow [69] | 128.13 |
| | LSGM [74] | 7.22 |
| | UDM [37] | 7.16 |
| | DDGAN [80] | 7.64 |
| | RGM [13] | **7.15** |
| **GAN** | PGGAN [34] | 8.03 |
| | Adv. LAE [59] | 19.2 |
| | VQ-GAN [16] | 10.2 |
| | DC-AE [55] | 15.8 |
| | StyleSwin [83] | **3.25** |
| **VAE** | NVAE [73] | 29.7 |
| | NCP-VAE [4] | 24.8 |
| | VAEBM [79] | **20.4** |
| **OT-based** | **UOTM**† | **6.36** |

Table 2: **Image Generation on CIFAR-10.**

| Class | Model | FID ($\downarrow$) | IS ($\uparrow$) |
|---|---|---|---|
| **GAN** | SNGAN+DGflow [5] | 9.62 | 9.35 |
| | AutoGAN [24] | 12.4 | 8.60 |
| | TransGAN [31] | 9.26 | 9.02 |
| | StyleGAN2 w/o ADA [35] | 8.32 | 9.18 |
| | StyleGAN2 w/ ADA [35] | 2.92 | **9.83** |
| | DDGAN (T=1)[80] | 16.68 | - |
| | DDGAN [80] | 3.75 | 9.63 |
| | RGM [13] | **2.47** | 9.68 |
| **Diffusion** | NCSN [68] | 25.3 | 8.87 |
| | DDPM [30] | 3.21 | 9.46 |
| | Score SDE (VE) [69] | 2.20 | 9.89 |
| | Score SDE (VP) [69] | 2.41 | 9.68 |
| | DDIM (50 steps) [67] | 4.67 | 8.78 |
| | CLD [15] | 2.25 | - |
| | Subspace Diffusion [32] | 2.17 | **9.94** |
| | LSGM [74] | **2.10** | 9.87 |
| **VAE&EBM** | NVAE [73] | 23.5 | 7.18 |
| | Glow [38] | 48.9 | 3.92 |
| | PixelCNN [75] | 65.9 | 4.60 |
| | VAEBM [79] | 12.2 | **8.43** |
| | Recovery EBM [22] | **9.58** | 8.30 |
| **OT-based** | WGAN [6] | 55.20 | - |
| | WGAN-GP[27] | 39.40 | 6.49 |
| | Robust-OT [8] | 21.57 | - |
| | AE-OT-GAN [3] | 17.10 | 7.35 |
| | OTM* (Small) [61] | 21.78 | - |
| | OTM (Large)† | 7.68 | 8.50 |
| | **UOTM** (Small)† | 12.86 | 7.21 |
| | **UOTM** (Large)† | **2.97**±0.07 | **9.68** |

[13], we refer to them as *large* model. Unless otherwise stated, we consider the *large* model. For implementation details of experiments, please refer to Appendix B.2.

## 5.1 Outlier Robustness of UOTM

One of the main features of UOT is its robustness against outliers. In this subsection, we investigate how this robustness is reflected in generative modeling tasks by comparing our UOTM and OT-based model (OTM). For evaluation, we generated two datasets that have 1-2% outliers: **Toy and Image datasets**. The toy dataset is a mixture of samples from $\mathcal{N}(1, 0.5^2)$ (in-distribution) and $\mathcal{N}(-1, 0.5^2)$ (outlier). Similarly, the image dataset is a mixture of CIFAR-10 [43] (in-distribution) and MNIST [14] (outlier) as in Balaji et al. [8].

Figure 2 illustrates the learned transport map $T$ and the probability density $T_\#\mu$ of generated distribution for OTM and UOTM models trained on the toy dataset. The density in Fig 2a demonstrates that OTM attempts to fit both the in-distribution and outlier samples simultaneously. However, this attempt causes undesirable behavior of transporting density outside the target support. In other words, in order to address 1-2% of outlier samples, *OTM generates additional failure samples that do not belong to the in-distribution or outlier distribution.* On the other hand, UOTM focuses on matching the in-distribution samples (Fig 2b). Moreover, the transport maps show that OTM faces challenging optimization. The right-hand side of Figure 2a and 2b visualize the correspondence between the source domain samples $x$ and the corresponding target domain samples $T(x)$. **Note that the optimal $T^\star$ should be a monotone-increasing function.** However, OTM failed to learn such a transport map $T$, while UOTM succeeded.

Figure 3 shows the generated samples from OTM and UOTM models on the image dataset. A similar phenomenon is observed in the image data. Some of the generative images from OTM show some unintentional artifacts. Also, MNIST-like generated images display a red or blue-tinted background. In contrast, UOTM primarily generates in-distribution samples. In practice, UTOM generates MNIST-like data at a very low probability of 0.2%. Notably, UOTM does not exhibit artifacts as in OTM. Furthermore, for quantitative comparison, we measured Fréchet Inception Distance (FID) score [29] by collecting CIFAR-10-like samples. Then, we compared this score with the model trained on a clean

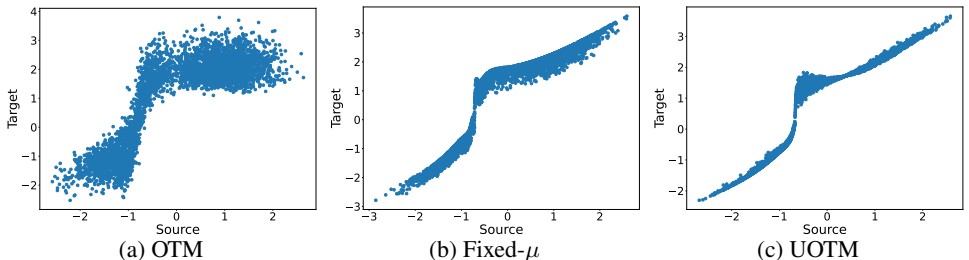

| (a) OTM | (b) Fixed-$\mu$ | (c) UOTM |

Figure 4: **Visualization of OT Map $(x, T(x))$ trained on clean Toy data**. The comparison suggests that Fixed-$\mu$ and UOTM models are closer to the optimal transport map compared to OTM.

dataset. The presence of outliers affected OTM by increasing FID score over 6 ($7.68 \rightarrow 13.82$) while the increase was less than 2 in UOTM ($2.97 \rightarrow 4.56$). In summary, the experiments demonstrate that UOTM is more robust to outliers.

## 5.2 UOTM as Generative Model

**Target Distribution Matching**   As aforementioned in Sec 3.3, UOT allows some flexibility to the marginal constraints of OT. This means that the optimal generator $T_\theta^\star$ does not necessarily transport the source distribution $\mu$ precisely to the target distribution $\nu$, i.e., $T_\# \mu \approx \nu$. However, the goal of the generative modeling task is to learn the target distribution $\nu$. In this regard, *we assessed whether our UOTM could accurately match the target (data) distribution.* Specifically, we measured KL divergence [78] between the generated distribution $T_\# \mu$ and data distribution $\nu$ on the toy dataset (See the Appendix B.1 for toy dataset details). Also, we employed the FID score for CIFAR-10 dataset, because FID assesses the Wasserstein distance between the generated samples and training data in the feature space of the Inception network [70].

In this experiment, our UOTM model is compared with two other methods: *OTM* [18] (Constraints in both marginals) and *UOTM with fixed* $\mu$ (Constraints only in the source distribution). We introduced *Fixed-$\mu$* variant because generative models usually sample directly from the input noise distribution $\mu$. This direct sampling implies the hard constraint on the source distribution. Note that this hard constraint corresponds to setting $\Psi_1^*(x) = x$ in Eq 9 (Remark 3.1). (See Appendix B.2 for implementation details of UOTM with fixed $\mu$.)

Table 1 shows the data distribution matching results. Interestingly, despite the soft constraint, **our UOTM matches data distribution better than OTM in both datasets.** In the toy dataset, UOTM achieves similar KL divergence with and without fixed $\mu$ for each $\tau$, which is much smaller than OTM. This result can be interpreted through the theoretical properties in Sec 3.3. Following Thm 3.3, both UOTM models exhibit smaller $D_{KL}(T_\# \mu | \nu)$ for the smaller $\tau$ in the toy dataset. It is worth noting that, unlike UOTM, $D_{KL}(T_\# \mu | \nu)$ does not change for OTM. This is because, in the standard OT problem, the optimal transport map $\pi^\star$ does not depend on $\tau$ in Eq 1. Moreover, in CIFAR-10, while Fixed-$\mu$ shows a similar FID score to OTM, UOTM significantly outperforms both models. As discussed in Thm 3.4, we interpret that relaxing both marginals improved the stability of the potential network $v$, which led to better convergence of the model on the more complex data. This convergence benefit can be observed in the learned transport map in Fig 4. Even on the toy dataset, the UTOM transport map presents a better-organized correspondence between source and target samples than OTM and Fixed-$\mu$. In summary, our model is competitive in matching the target distribution for generative modeling tasks.

**Image Generation**   We evaluated our UOTM model on the two generative model benchmarks: CIFAR-10 [43] ($32 \times 32$) and CelebA-HQ [46] ($256 \times 256$). For quantitative comparison, we adopted FID [29] and Inception Score (IS) [62]. The qualitative performance of UOTM is depicted in Fig 1. As shown in Table 2, UOTM model demonstrates state-of-the-art results among existing OT-based methods, with an FID of 2.97 and IS of 9.68. Our UOTM outperforms the second-best performing OT-based model OTM(Large), which achieves an FID of 7.68, by a large margin. Moreover, UOTM(Small) model surpasses another UOT-based model with the same backbone network (Robust-OT), which achieves an FID of 21.57. Furthermore, our model achieves the state-of-the-art FID score of 6.36 on CelebA-HQ ($256 \times 256$). To the best of our knowledge, UOTM is the first OT-based generative model that has shown comparable results with state-of-the-art models in various datasets.

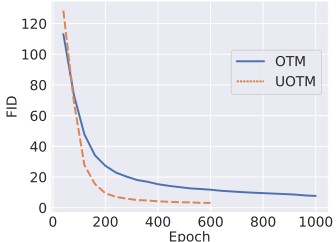 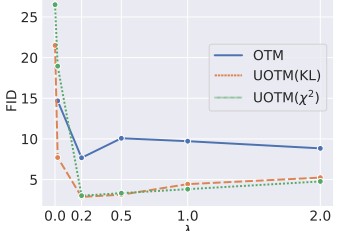 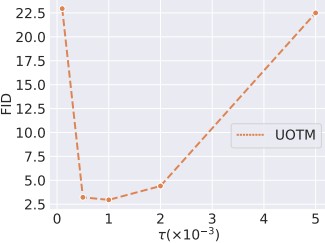

Figure 5: **FID Scores during Training on CIFAR-10.**

Figure 6: **Ablation Study on Regularizer Intensity** $\lambda$.

Figure 7: **Ablation Study on** $\tau$ **in** $c(x,y) = \tau\|x - y\|_2^2$.

## 5.3 Fast Convergence

The discussion in Thm 3.4 suggests that UOTM offers some optimization benefits over OTM, such as faster and more stable convergence. In addition to the transport map visualization in the toy dataset (Fig 2 and 4), we investigate the faster convergence of UOTM on the image dataset. In CIFAR-10, UOTM converges in 600 epochs, whereas OTM takes about 1000 epochs (Fig 5). To achieve the same performance as OTM, UOTM needs only 200 epochs of training, which is about five times faster. In addition, compared to several approaches that train CIFAR-10 with NCSN++ (*large model*) [69], our model has the advantage in training speed; Score SDE [69] takes more than 70 hours for training CIFAR-10, 48 hours for DDGAN [80], and 35-40 hours for RGM on four Tesla V100 GPUs. OTM takes approximately 30-35 hours to converge, while our model only takes about 25 hours.

## 5.4 Ablation Studies

**Generalized** $\Psi_1$, $\Psi_2$ We investigate the various choices of Csiszàr divergence in UOT problem, i.e., $\Psi_1^*$ and $\Psi_2^*$ in Eq 9. As discussed in Remark 3.2, the necessary condition of feasible $\Psi^*$ is non-decreasing, convex, and differentiable functions. For instance, setting the Csiszàr divergence $D_\Psi$ as $f$-divergence families such as KL divergence, $\chi^2$ divergence, and Jensen-Shannon divergence satisfies the aforementioned conditions. For practical optimization, we chose $\Psi^*(x)$ that gives finite values for all $x$, such as KL divergence (Eq 13) and $\chi^2$ divergence (Eq 14):

Table 4: **Ablation Study on Csiszàr Divergence** $D_{\Psi_i}(\cdot|\cdot)$.

| $(\Psi_1, \Psi_2)$ | FID |
|---|---|
| (KL,KL) | 2.97 |
| $(\chi^2, \chi^2)$ | 3.02 |
| (KL, $\chi^2$) | 3.21 |
| $(\chi^2, $ KL$)$ | 2.78 |
| (Softplus, Softplus) | 3.17 |

$$\Psi(x) = \begin{cases} (x-1)^2, & \text{if } x \geq 0 \\ \infty, & \text{if } x < 0 \end{cases}, \qquad \Psi^*(x) = \begin{cases} \frac{1}{4}x^2 + x, & \text{if } x \geq -2 \\ -1, & \text{if } x < -2 \end{cases}. \tag{14}$$

We assessed the performance of our UOTM models for the cases where $D_\Psi$ is KL divergence or $\chi^2$ divergence. Additionally, we tested our model when $\Psi^* = \text{Softplus}$ (Eq 37), which is a direct parametrization of $\Psi^*$. Table 4 shows that our UOTM model achieves competitive performance across all five combinations of $(\Psi_1^*, \Psi_2^*)$.

$\lambda$ **in Regularizer** $\mathcal{L}_{reg}$ We conduct an ablation study to investigate the effect of the regularization term $\mathcal{L}_{reg}$ on our model's performance. The WGAN family is known to exhibit unstable training dynamics and to highly depend on the regularization term [6, 27, 51, 56]. Similarly, Figure 6 shows that OTM is also sensitive to the regularization term and fails to converge without it (OTM shows an FID score of 152 without regularization. Hence, we excluded this result in Fig 6 for better visualization). In contrast, our model is robust to changes in the regularization hyperparameter and produces reasonable performance even without the regularization.

$\tau$ **in the Cost Function** We performed an ablation study on $\tau$ in the cost function $c(x,y) = \tau\|x - y\|_2^2$. In our model, the hyperparameter $\tau$ controls the relative weight between the cost $c(x,y)$ and the marginal matching $D_{\Psi_1}, D_{\Psi_2}$ in Eq 4. In Fig 7, our model maintains a decent performance of FID($\leq 5$) for $\tau = \{0.5, 1, 2\} \times 10^{-3}$. However, the FID score sharply degrades at $\tau = \{0.1, 5\} \times 10^{-3}$. Thm 3.3 suggests that a smaller $\tau$ leads to a tighter distribution matching. However, this cannot explain the degradation at $\tau = 0.1 \times 10^{-3}$. We interpret this is because the cost function provides some regularization effect that prevents a mode collapse of the model (See the Appendix C.2 for a detailed discussion).

# 6 Conclusion

In this paper, we proposed a generative model based on the semi-dual form of Unbalanced Optimal Transport, called UOTM. Our experiments demonstrated that UOTM achieves better target distribution matching and faster convergence than the OT-based method. Moreover, UOTM outperformed existing OT-based generative model benchmarks, such as CIFAR-10 and CelebA-HQ-256. The potential negative societal impact of our work is that the Generative Model often learns the dependence in the semantics of data, including any existing biases. Hence, deploying a Generative Model in real-world applications requires careful monitoring to prevent the amplification of existing societal biases present in the data. It is important to carefully control the training data and modeling process of Generative Models to mitigate potential negative societal impacts.

## Acknowledgements

This work was supported by KIAS Individual Grant [AP087501] via the Center for AI and Natural Sciences at Korea Institute for Advanced Study, the NRF grant [2012R1A2C3010887] and the MSIT/IITP ([1711117093], [2021-0-00077], [No. 2021-0-01343, Artificial Intelligence Graduate School Program(SNU)]).

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

# A  Proofs

**Notations and Assumptions**  Let $\mathcal{X}$ and $\mathcal{Y}$ be compact complete metric spaces which are subsets of $\mathbb{R}^d$, and $\mu$, $\nu$ be positive Radon measures of the mass 1. Let $\Psi_1$ and $\Psi_2$ be a convex, differentiable, nonnegative function defined on $\mathbb{R}$. We assume that $\Psi_1(0) = \Psi_2(0) = 1$, and $\Psi_1(x) = \Psi_2(x) = \infty$ for $x < 0$. The convex conjugate of $\Psi$ is denoted by $\Psi^*$. We assume that $\Psi_1^*$ and $\Psi_2^*$ are differentiable and non-decreasing on its domain. Moreover, let $c$ be a quadratic cost, $c(x, y) = \tau \|x - y\|_2^2$, where $\tau$ is positive constant.

First, for completeness, we present the derivation of the dual and semi-dual formulations of the UOT problem.

**Theorem A.1** ([12, 21, 72]). *The dual formulation of Unbalanced OT (4) is given by*

$$C_{ub}(\mu, \nu) = \sup_{u(x)+v(y)\leq c(x,y)} \left[ \int_{\mathcal{X}} -\Psi_1^*(-u(x))d\mu(x) + \int_{\mathcal{Y}} -\Psi_2^*(-v(y))d\nu(y) \right], \quad (15)$$

*where $u \in \mathcal{C}(\mathcal{X})$, $v \in \mathcal{C}(\mathcal{Y})$ with $u(x) + v(y) \leq c(x, y)$. Note that strong duality holds.*

*Also, the semi-dual formulation of Unbalanced OT (4) is given by*

$$C_{ub}(\mu, \nu) = \sup_{v\in\mathcal{C}} \left[ \int_{\mathcal{X}} -\Psi_1^*\left(-v^c(x)\right)) d\mu(x) + \int_{\mathcal{Y}} -\Psi_2^*(-v(y))d\nu(y) \right], \quad (16)$$

*where $v^c(x)$ denotes the c-transform of $v$, i.e., $v^c(x) = \inf_{y\in\mathcal{Y}} (c(x, y) - v(y))$.*

*Proof.* The primal problem Eq 4 can be rewritten as

$$\inf_{\pi\in\mathcal{M}_+(\mathcal{X}\times\mathcal{Y})} \int_{\mathcal{X}\times\mathcal{Y}} c(x, y)d\pi(x, y) + \int_{\mathcal{X}} \Psi_1\left(\frac{\pi_0(x)}{d\mu(x)}\right) d\mu(x) + \int_{\mathcal{Y}} \Psi_2\left(\frac{\pi_1(y)}{d\nu(y)}\right) d\nu(y). \quad (17)$$

We introduce slack variables $P$ and $Q$ such that $P = \pi_0$ and $Q = \pi_1$. By Lagrange multipliers where $u$ and $v$ associated to the constraints, the Lagrangian form of Eq 17 is

$$\mathcal{L}(\pi, P, Q, u, v) := \int_{\mathcal{X}\times\mathcal{Y}} c(x, y)d\pi(x, y) \quad (18)$$

$$+ \int_{\mathcal{X}} \Psi_1\left(\frac{dP(x)}{d\mu(x)}\right) d\mu(x) + \int_{\mathcal{Y}} \Psi_2\left(\frac{dQ(y)}{d\nu(y)}\right) d\nu(y) \quad (19)$$

$$+ \int_{\mathcal{X}} u(x)\left(dP(x) - \int_{\mathcal{Y}} d\pi(x, y)\right) + \int_{\mathcal{Y}} v(y)\left(dQ(y) - \int_{\mathcal{X}} d\pi(x, y)\right). \quad (20)$$

The dual Lagrangian function is $g(u, v) = \inf_{\pi, P, Q} \mathcal{L}(\pi, P, Q, u, v)$. Then, $g(u, v)$ reduces into three distinct minimization problems for each variables;

$$g(u, v) = \inf_{P} \left[ \int_{X} u(x)dP(x) + \Psi_1\left(\frac{dP(x)}{d\mu(x)}\right) d\mu(x) \right]$$

$$+ \inf_{Q} \left[ \int_{Y} v(y)dQ(y) + \Psi_2\left(\frac{dQ(y)}{d\nu(y)}\right) d\nu(y) \right] \quad (21)$$

$$+ \inf_{\pi} \left[ \int_{\mathcal{X}\times\mathcal{Y}} (c(x, y) - u(x) - v(y)) d\pi(x, y) \right].$$

Note that the first term of Eq 21 is

$$-\sup_{P} \left[ \int_{\mathcal{X}} \left[ -u(x)\frac{dP(x)}{d\mu(x)} - \Psi_1\left(\frac{dP(x)}{d\mu(x)}\right) \right] d\mu(x) \right] = -\int_{\mathcal{X}} \Psi_1^*\left(-u(x)\right) d\mu(x). \quad (22)$$

Thus, Eq 21 can be rewritten as follows:

$$g(u, v) = \inf_{\pi} \left[ \int_{\mathcal{X}\times\mathcal{Y}} (c(x, y) - u(x) - v(y)) d\pi(x, y) \right]$$

$$- \int_{\mathcal{X}} \Psi_1^*\left(-u(x)\right) d\mu(x) - \int_{\mathcal{Y}} \Psi_2^*\left(-v(x)\right) d\nu(x). \quad (23)$$

Suppose that $c(x, y) - u(x) - v(y) < 0$ for some point $(x, y)$. Then, by concentrating all mass of $\pi$ into the point, the value of the minimization problem of Eq 23 becomes $-\infty$. Thus, to avoid the trivial solution, we should set $c(x, y) - u(x) - v(y) \geq 0$ almost everywhere. Within the constraint, the solution of the optimization problem of Eq 23 is $c(x, y) = u(x) + v(y)$ $\pi$-almost everywhere. Finally, we obtain the dual formulation of UOT:

$$\sup_{u(x)+v(y)\leq c(x,y)} \left[ \int_{\mathcal{X}} -\Psi_1^*(-u(x))d\mu(x) + \int_{\mathcal{Y}} -\Psi_2^*(-v(y))d\nu(y) \right], \tag{24}$$

where $c(x, y) = u(x) + v(y)$ $\pi$-almost everywhere (since $\Psi_1^*$ and $\Psi_2^*$ is non-decreasing). In other words,

$$\sup_{(u,v)\in\mathcal{C}(\mathcal{X})\times\mathcal{C}(\mathcal{Y})} \left[ \int_{\mathcal{X}} -\Psi_1^*(-u(x))d\mu(x) + \int_{\mathcal{Y}} -\Psi_2^*(-v(y))d\nu(y) - \imath(u + v \leq c) \right], \tag{25}$$

where $\imath$ is a convex indicator function. Note that we assume $\Psi_1^*$ and $\Psi_2^*$ are convex, non-decreasing, and differentiable. Since $c \geq 0$, by letting $u \equiv -1$ and $v \equiv -1$, we can easily see that all three terms in Eq 25 are finite values. Thus, we can apply Fenchel-Rockafellar's theorem, which implies that the strong duality holds. Finally, combining the constraint $u(x) + v(y) \leq c(x, y)$ with tightness condition, i.e. $c(x, y) = u(x) + v(y)$ $\pi$-almost everywhere, we obtain the following semi-dual form:

$$\sup_{v\in\mathcal{C}} \left[ \int_{\mathcal{X}} -\Psi_1^* \left( -\inf_{y\in\mathcal{Y}}[c(x, y) - v(y)] \right) \mu(x) + \int_{\mathcal{Y}} -\Psi_2^*(-v(y))\nu(y) \right]. \tag{26}$$

$\square$

Now, we state precisely and prove Theorem 3.3.

**Lemma A.2.** *For any $(u, v) \in \mathcal{C}(\mathcal{X}) \times \mathcal{C}(\mathcal{Y})$, let*

$$I(u, v) := \int_{\mathcal{X}} -\Psi_1^*(-u(x))d\mu(x) + \int_{\mathcal{Y}} -\Psi_2^*(-v(y))d\nu(y). \tag{27}$$

*Then, under the assumptions in Appendix A, the functional derivative of $I(u, v)$ is*

$$\int_{\mathcal{X}} \delta u(x)\Psi_1^{*\prime}(-u(x))d\mu(x) + \int_{\mathcal{Y}} \delta v(y)\Psi_2^{*\prime}(-v(y))d\nu(y), \tag{28}$$

*where where $(\delta u, \delta v) \in \mathcal{C}(\mathcal{X}) \times \mathcal{C}(\mathcal{Y})$ denote the variations of $(u, v)$, respectively.*

*Proof.* For any $(f, g) \in \mathcal{C}(\mathcal{X}) \times \mathcal{C}(\mathcal{Y})$, let

$$I(f, g) = \int_{\mathcal{X}} -\Psi_1^*(-f(x))d\mu(x) + \int_{\mathcal{Y}} -\Psi_2^*(-g(y))d\nu(y).$$

Then, for arbitrarily given potentials $(u, v) \in \mathcal{C}(\mathcal{X}) \times \mathcal{C}(\mathcal{Y})$, and perturbation $(\eta, \zeta) \in \mathcal{C}(\mathcal{X}) \times \mathcal{C}(\mathcal{Y})$ which satisfies $u(x) + \eta(x) + v(y) + \zeta(y) \leq c(x, y)$,

$$\mathcal{L}(\epsilon) := \frac{d}{d\epsilon} I(u + \epsilon\eta, v + \epsilon\zeta) = \int_{\mathcal{X}} \eta\Psi_1^{*\prime}(-((u + \epsilon\eta)(x)))d\mu(x) + \int_{\mathcal{Y}} \zeta\Psi_2^{*\prime}(-((v + \epsilon\zeta)(y)))d\nu(y).$$

Note that $\mathcal{L}(\epsilon)$ is a continuous function. Thus, the functional derivative of $I(u, v)$, i.e. $\mathcal{L}(0)$, is given by calculus of variation (Chapter 8, Evans [17])

$$\int_{\mathcal{X}} \delta u(x)\Psi_1^{*\prime}(-u(x))d\mu(x) + \int_{\mathcal{Y}} \delta v(y)\Psi_2^{*\prime}(-v(y))d\nu(y), \tag{29}$$

where $(\delta u, \delta v) \in \mathcal{C}(\mathcal{X}) \times \mathcal{C}(\mathcal{Y})$ denote the variations of $(u, v)$, respectively. $\square$

**Theorem A.3.** *Suppose that $\mu$ and $\nu$ are probability densities defined on $\mathcal{X}$ and $\mathcal{Y}$. Given the assumptions in Appendix A, suppose that $\mu, \nu$ are absolutely continuous with respect to Lebesgue measure and $\Psi^*$ is continuously differentiable. Assuming that the optimal potential $v^\star = \inf_{v\in\mathcal{C}} J(v)$ exists, $v^\star$ is a solution of the following objective*

$$\tilde{J}(v) = \int_{\mathcal{X}} -v^c(x)d\tilde{\mu}(x) + \int_{\mathcal{Y}} -v(y)d\tilde{\nu}(y), \tag{30}$$

where $\tilde{\mu}(x) = \Psi_1^{*'}(-v^{\star c}(x))\mu(x)$ and $\tilde{\nu}(y) = \Psi_2^{*'}(-v^{\star}(y))\nu(y)$. *Note that the assumptions guarantee the existence of optimal transport map $T^{\star}$ between $\tilde{\mu}$ and $\tilde{\nu}$. Furthermore, $T^{\star}$ satisfies*

$$T^{\star}(x) \in \operatorname{arginf}_{y \in \mathcal{Y}} \left[ c(x, y) - v^{\star}(y) \right], \tag{31}$$

$\mu$-*a.s..In particular,* $D_{\Psi_1}(\tilde{\mu}|\mu) + D_{\Psi_2}(\tilde{\nu}|\nu) \leq \tau \mathcal{W}_2^2(\mu, \nu)$ *where* $\mathcal{W}_2(\mu, \nu)$ *is a Wasserstein-2 distance between $\mu$ and $\nu$.*

*Proof.* By Lemma A, derivative of Eq 27 is

$$\int_{\mathcal{X}} \delta u(x) \Psi_1^{*'}(-u(x)) d\mu(x) + \int_{\mathcal{Y}} \delta v(y) \Psi_2^{*'}(-v(y)) d\nu(y), \tag{32}$$

Note that such linearization must satisfy the inequality constraint $u(x) + \delta u(x) + v(y) + \delta v(y) \leq c(x, y)$ to give admissible variations. Then, the linearized dual form can be re-written as follows:

$$\sup_{(u,v) \in \mathcal{C}(\mathcal{X}) \times \mathcal{C}(\mathcal{Y})} \int_X \delta u(x) d\tilde{\mu}(x) + \int_{\mathcal{Y}} \delta v(y) d\tilde{\nu}(y), \tag{33}$$

where $\delta u(x) + \delta v(y) \leq c(x, y) - u(x) - v(y)$, $\tilde{\mu}(x) = \Psi_1^{*'}(-u(x))\mu(x)$, and $\tilde{\nu}(y) = \Psi_2^{*'}(-v(y))\nu(y)$. Thus, whenever $(u^{\star}, v^{\star})$ is optimal, the linearized dual problem is non-positive due to the inequality constraint and tightness (i.e. $c(x, y) - u^{\star}(x) - v^{\star}(y) = 0$ $\pi$-almost everywhere), which implies that Eq 33 achieves its optimum at $(u^{\star}, v^{\star})$. Note that Eq 33 is a linearized dual problem of the following dual problem:

$$\sup_{u(x)+v(y) \leq c(x,y)} \left[ \int_{\mathcal{X}} u(x) d\tilde{\mu}(x) + \int_{\mathcal{Y}} v(y) d\tilde{\nu}(y) \right]. \tag{34}$$

Hence, the UOT problem can be reduced into standard OT formulation with marginals $\tilde{\mu}$ and $\tilde{\nu}$. Moreover, the continuity of $\Psi^{*'}$ gives that $\tilde{\mu}, \tilde{\nu}$ are absolutely continuous with respect to Lebesgue measure and have a finite second moment (Note that $\mathcal{X}, \mathcal{Y}$ are compact. Hence, $\Psi^{*'}$ is bounded on $\mathcal{X}, \mathcal{Y}$, and $\mathcal{X}, \mathcal{Y}$ are bounded). By Theorem 2.12 in Villani [76], there exists a unique measurable OT map $T^{\star}$ which solves Monge problem between $\tilde{\mu}$ and $\tilde{\nu}$. Furthermore, as shown in Remark 5.13 in Villani et al. [77], $T^{\star}(x) \in \operatorname{arginf}_{y \in \mathcal{Y}} \left[ c(x, y) - v(y) \right]$.

Finally, if we constrain $\pi$ into $\Pi(\mu, \nu)$, then the solution of Eq 4 is $C_{ub} = \tau \mathcal{W}_2^2(\mu, \nu)$. Thus, for the optimal $\pi \in \mathcal{M}_+$ where $\pi_0 = \tilde{\mu}$ and $\pi_1 = \tilde{\nu}$, it is trivial that

$$\tau \mathcal{W}_2^2(\mu, \nu) \geq \tau \mathcal{W}_2^2(\tilde{\mu}, \tilde{\nu}) + D_{\Psi_1}(\tilde{\mu}|\mu) + D_{\Psi_2}(\tilde{\nu}|\nu), \tag{35}$$

which proves the last statement of the theorem. $\square$

**Theorem A.4** ([21]). *$J$ is convex. Suppose $v$ is a $\lambda$-strongly convex, $v^c$ and $v$ are uniformly bounded on the support of $\mu$ and $\nu$, respectively, and $\Psi_1^*, \Psi_2^*$ are strongly convex on every compact set. Then, it holds*

$$J(v) - J(v^{\star}) \geq \frac{1}{2\lambda} \mathbb{E}_{\tilde{\mu}} \left[ \| \nabla (v^c - v^{\star c}) \|_2^2 \right] + C_1 \mathbb{E}_{\mu} \left[ (v^c - v^{\star c})^2 \right] + C_2 \mathbb{E}_{\nu} \left[ (v - v^{\star})^2 \right], \tag{36}$$

*for some positive constant $C_1$ and $C_2$. Furthermore, $\tilde{J}(v) - \tilde{J}(v^{\star}) \geq \frac{1}{2\lambda} \mathbb{E}_{\tilde{\mu}} \left[ \| \nabla (v^c - v^{\star c}) \|_2^2 \right]$.*

*Proof.* See [21]. $\square$

# B Implementation Details

## B.1 Data

Unless otherwise stated, **the source distribution $\mu$ is a standard Gaussian distribution $\mathcal{N}(\mathbf{0}, \mathbf{I})$** with the same dimension as the target distribution $\nu$.

**Toy Data** We generated 4000 samples for each Toy dataset: Toy dataset for Outlier Robustness (Sec 5.1) and Toy dataset for Target Distribution Matching (Sec 5.2). In Fig 2 of Sec 5.1, the target dataset consists of 99% of samples from $\mathcal{N}(1, 0.5^2)$ and 1% of samples from $\mathcal{N}(-1, 0.5^2)$. In Tab 1 and Fig 4 of Sec 5.2, the source data is composed of 50% of samples from $\mathcal{N}(-1, 0.5^2)$ and 50% of samples from $\mathcal{N}(1, 0.5)$. The target data is sampled from the mixture of two Gaussians: $\mathcal{N}(-1, 0.5)$ with a probability of $1/3$ and $\mathcal{N}(2, 0.5)$ with a probability of $2/3$.

**CIFAR-10**  We utilized all 50,000 samples. For each image $x$, we applied random horizontal flip augmentation of probability 0.5 and transformed it by $2x - 1$ to scale the values within the $[-1, 1]$ range. In the outlier experiment (Fig 3 of Sec 5.1), we added additional 500 MNIST samples to the clean CIFAR-10 dataset. When adding additional MNIST samples, each sample was resized to $3 \times 32 \times 32$ by duplicating its channel to create a 3-channeled image. Then, the samples were transformed by $2x - 1$. For the image generation task in Sec 5.2, we followed the source distribution embedding of Rout et al. [61] for the *small* model, i.e., the Gaussian distribution $\mu$ with the size of $3 \times 8 \times 8$ is bicubically upsampled to $3 \times 32 \times 32$.

**CelebA-HQ**  We used all 120,000 samples. For each image $x$, we resized it to $256 \times 256$ and applied random horizontal flip augmentation with a probability of 0.5. Then, we linearly transformed each image by $2x - 1$ to scale the values within $[-1, 1]$.

### B.2   Implementation details

**Toy data**  For all the Toy dataset experiments, we used the same generator and discriminator architectures. The dimension of the auxiliary variable $z$ is set to one. For a generator, we passed $z$ through two fully connected (FC) layers with a hidden dimension of 128, resulting in 128-dimensional embedding. We also embedded data $x$ into the 128-dimensional vector by passing it through three-layered ResidualBlock [68]. Then, we summed up the two vectors and fed it to the final output module. The output module consists of two FC layers. For the discriminator, we used three layers of ResidualBlock and two FC layers (for the output module). The hidden dimension is 128. Note that the SiLU activation function is used. We used a batch size of 256, a learning rate of $10^{-4}$, and 2000 epochs. Like WGAN [6], we used the number of iterations of the discriminator per generator iteration as five. For OTM experiments, since OTM does not converge without regularization, we set $R_1$ regularization to $\lambda = 0.01$.

**CIFAR-10**  For the *small* model setting of UOTM, we employed the architecture of Balaji et al. [8]. Note that this is the same model architecture as in Rout et al. [61]. We set a batch size of 128, 200 epochs, a learning rate of $2 \times 10^{-4}$ and $10^{-4}$ for the generator and discriminator, respectively. Adam optimizer with $\beta_1 = 0.5$, $\beta_2 = 0.9$ is employed. Moreover, we use $R_1$ regularization of $\lambda = 0.2$. For the *large* model setting of UOTM, we followed the implementation of Choi et al. [13] unless stated. We trained for 600 epochs with $\lambda = 0.2$ for every iteration. Moreover, in OTM (Large) experiment, we trained the model for 1000 epochs. The other hyperparameters are the same as UOTM. Tt For the ablation studies in Sec 5.4, we reported the best FID score until 600 epoch for UOTM and 1000 epoch for OTM because the optimal epoch differs for different hyperparameters. Additionally, for the ablation study on $\tau$ in the cost function (Fig 7), we reported the best FID score among three $R_1$ regularizer parameters: $\lambda = 0.2, 0.02, 1$. To ensure the reliability of our experiments, we conducted three experiments with different random seeds and presented the mean and standard deviation in Tab 2 (UOTM (Large)).

**CelebA-HQ** $(256 \times 256)$  We set $\tau = 10^{-5}$ and used $R_1$ regularizer with $\lambda = 5$ for every iteration. The model is trained for 450 epochs with an exponential moving average of 0.999. The other network architecture and optimizer settings are the same as Xiao et al. [80] and Choi et al. [13].

**Evaluation Metric**  We used the implementation of Wang et al. [78] for the KL divergence in Tab 1 of Sec 5.2. We set $k = 2$ (See Wang et al. [78]). For the evaluation of image datasets, we used 50,000 generated samples to measure IS and FID scores.

**Ablation Study on Csiszàr Divergence**  To ensure self-containedness, we include the definition of Softplus function employed in Table 4.

$$\text{Softplus}(x) = \log(1 + \exp(x)) \tag{37}$$

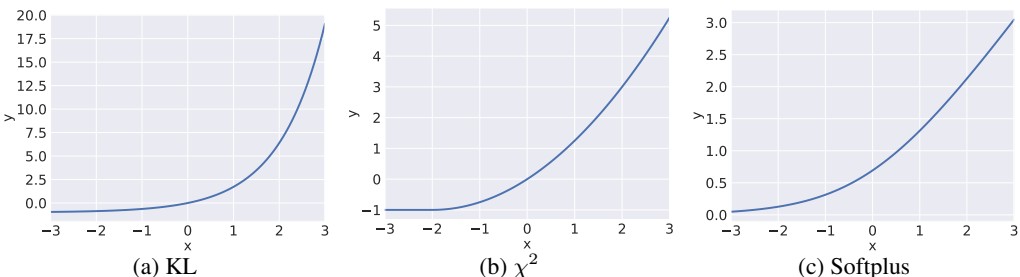

Figure 8: Various types of $\Psi^*$.

## C  Additional Discussion on UOTM

### C.1  Stability of UOTM

In this section, we provide an intuitive explanation for the stability and convergence of our model. Note that our model employs a non-decreasing, differentiable, and convex $\Psi^*$ (Sec 3) as depicted in Fig 8. In contrast, OTM corresponds to the specific case of $\Psi^*(x) = x$. Now, recall our objective function for the potential $v_\phi$ and the generator $T_\theta$:

$$\mathcal{L}_v = \frac{1}{|X|} \sum_{x \in X} \Psi_1^* \left( -c\left(x, T_\theta(x, z)\right) + v_\phi\left(T_\theta(x, z)\right) \right) + \frac{1}{|Y|} \sum_{y \in Y} \Psi_2^*(-v_\phi(y)), \qquad (38)$$

$$\mathcal{L}_T = \frac{1}{|X|} \sum_{x \in X} \left( c\left(x, T_\theta(x, z)\right) - v_\phi(T_\theta(x, z)) \right). \qquad (39)$$

Then, the gradient descent step of potential $v_\phi$ for a single real data $y$ with a learning rate of $\gamma$ can be expressed as follows:

$$\phi - \gamma \nabla_\phi \mathcal{L}_{v_\phi} = \phi - \gamma \nabla_\phi \Psi_2^*(-v_\phi(y)) = \phi + \gamma \Psi_2^{*\prime}(-v_\phi(y)) \nabla_\phi v_\phi(y). \qquad (40)$$

Suppose that our potential network $v_\phi$ assigns a low potential for $y$. Since the objective of the potential network is to assign high values to the real data, this indicates that $v_\phi$ fails to allocate the appropriate potential to the real data point $y$. Then, because $-v_\phi(y)$ is large, $\Psi^{*\prime}(-v_\phi(y))$ in Eq 40 is large, as shown in Fig 8. In other words, the potential network $v_\phi$ takes a stronger gradient step on the real data $y$ where $v_\phi$ fails. Note that this does not happen for OTM because $\Psi^{*\prime}(x) = 1$ for all $x$. In this respect, *UOTM enables adaptive updates for each sample, with weaker updates on well-performing data and stronger updates on poor-performing data.* We believe that UOTM achieves significant performance improvements over OTM by this property. Furthermore, UOTM attains higher stability because the smaller updates on well-behaved data prevent the blow-up of the model output.

### C.2  Qualitative Comparison on Varying $\tau$

In the ablation study on $\tau$, UOTM exhibited a similar degradation in FID score($\geq 20$) at $\tau(\times 10^{-3}) = 0.1, 5$. However, the generated samples are significantly different. Figure 9 illustrates the generated samples from trained UOTM for the cases when $\tau$ is too small ($\tau = 0.1$) and when $\tau$ is too large $\tau = 5$. When $\tau$ is too small (Fig 9 (left)), the generated images show repetitive patterns. This lack of diversity suggests mode collapse. We consider that the cost function $c$ provides some regularization effect preventing mode collapse. For the source sample (noise) $x$, the cost function is defined as $c(x, T(x)) = \tau \|x - T(x)\|_2^2$. Hence, minimizing this cost function encourages the generated images $T(x)$ to disperse by aligning each image to the corresponding noise $x$. Therefore, increasing $\tau$ from 0.1 led to better FID results in Fig 7.

On the other hand, when $\tau$ is too large Fig 9 (right), the generative samples tend to be noisy. Because the cost function is $c(x, T(x)) = \tau \|x - T(x)\|_2^2$, too large $\tau$ induces the generated image $T(x)$ to be similar to the noise $x$. This resulted in the degradation of the FID score for $\tau = 5$ in Fig 7. As discussed in Sec 5.4, this phenomenon can also be understood through Thm 3.3.

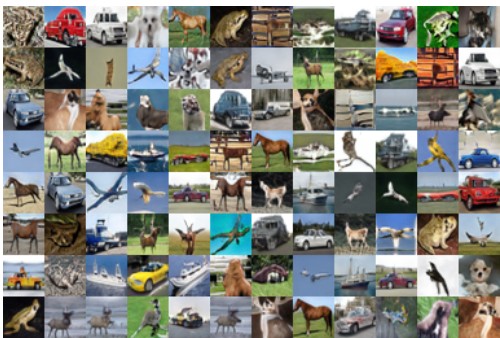 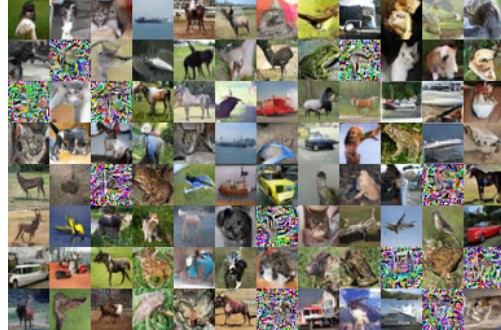

Figure 9: Generated samples from UOTM trained on CIFAR-10 with **Left**: $\tau = 0.1(\times 10^{-3})$ and **Right**: $\tau = 5(\times 10^{-3})$.

# D  Additional Results

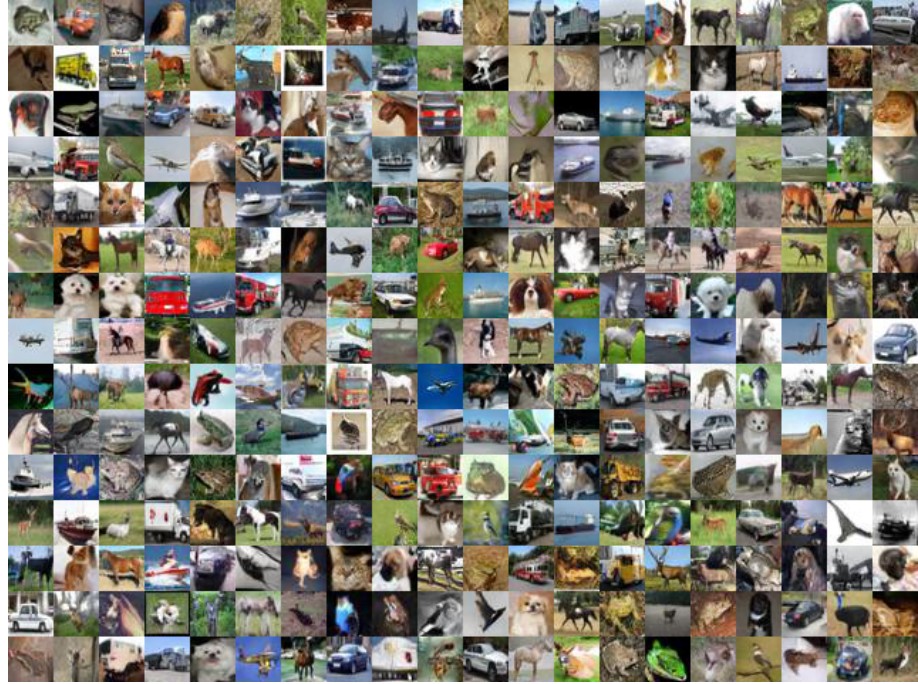

Figure 10: Generated samples from UOTM trained on CIFAR10 for $(\Psi_1, \Psi_2) = (KL, KL)$.

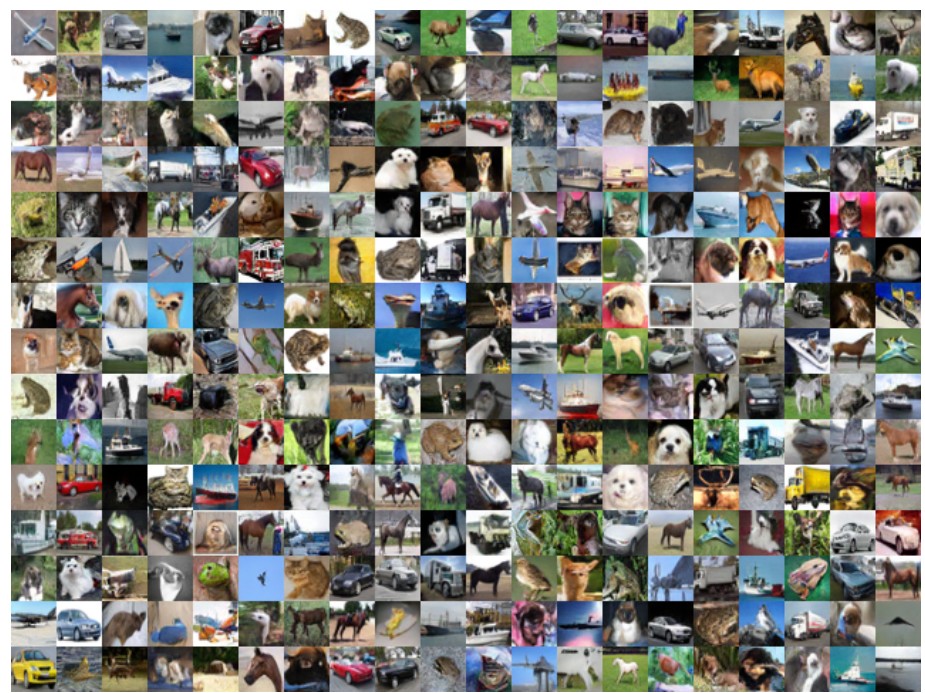

Figure 11: Generated samples from UOTM trained on CIFAR10 for $(\Psi_1, \Psi_2) = (\chi^2, \chi^2)$.

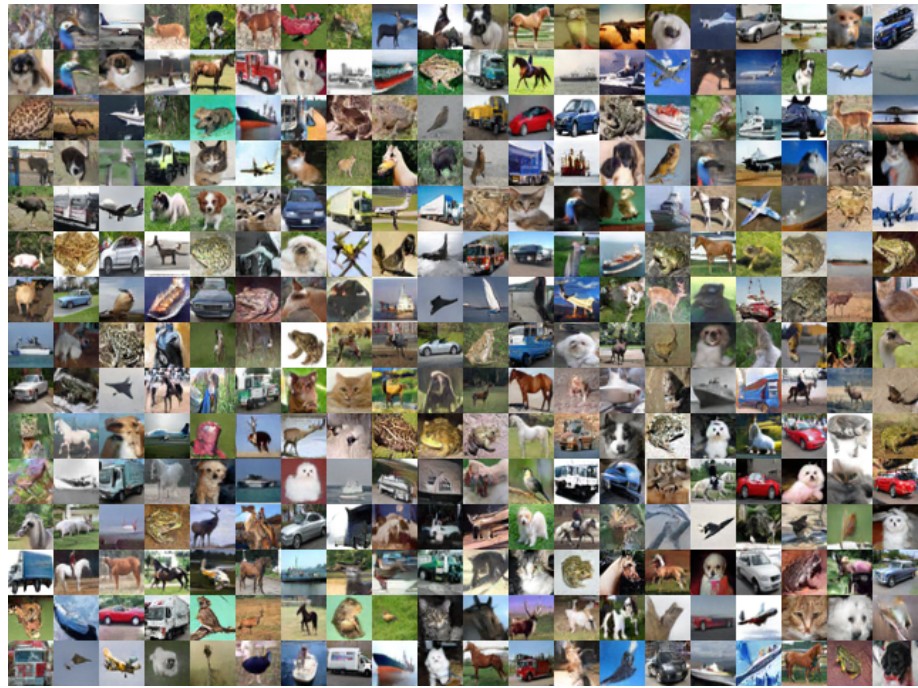

Figure 12: Generated samples from UOTM trained on CIFAR10 for $(\Psi_1, \Psi_2) = (KL, \chi^2)$.

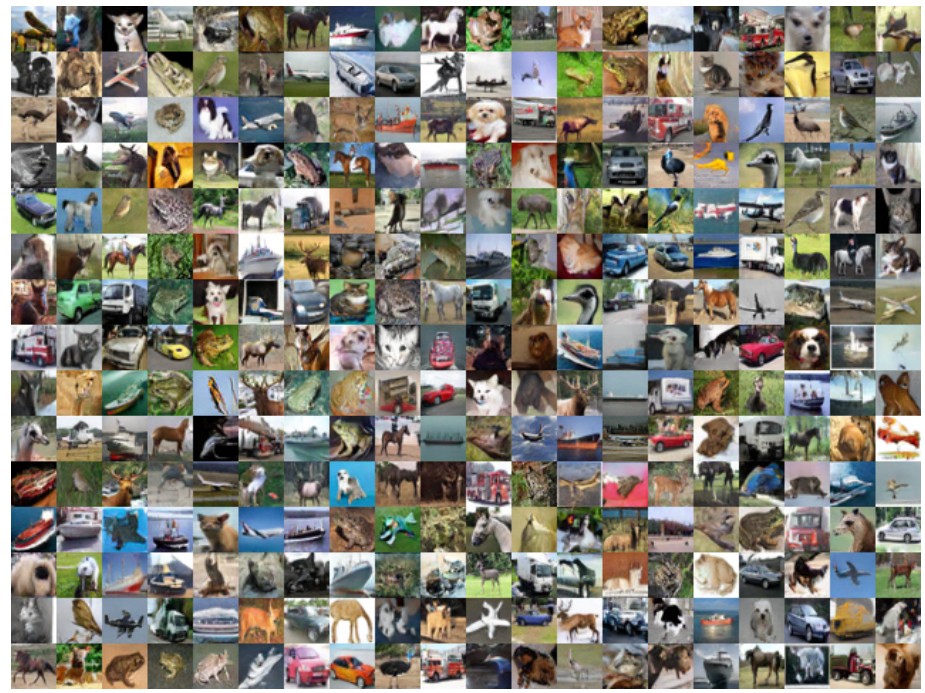

Figure 13: Generated samples from UOTM trained on CIFAR10 for $(\Psi_1, \Psi_2) = (\chi^2, KL)$.

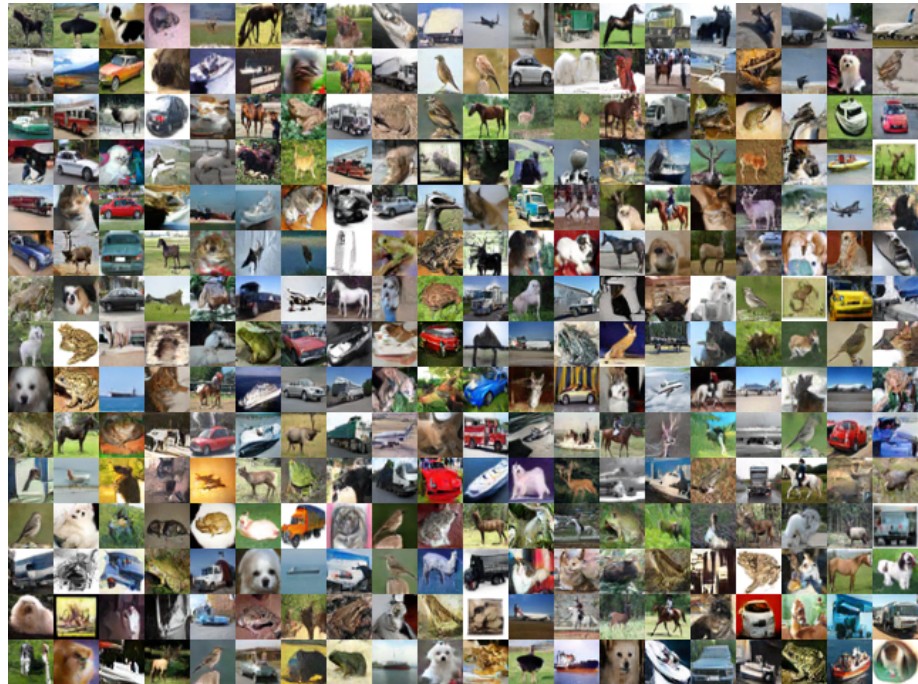

Figure 14: Generated samples from UOTM trained on CIFAR10 for $(\Psi_1, \Psi_2) = (\text{Softplus}, \text{Softplus})$.

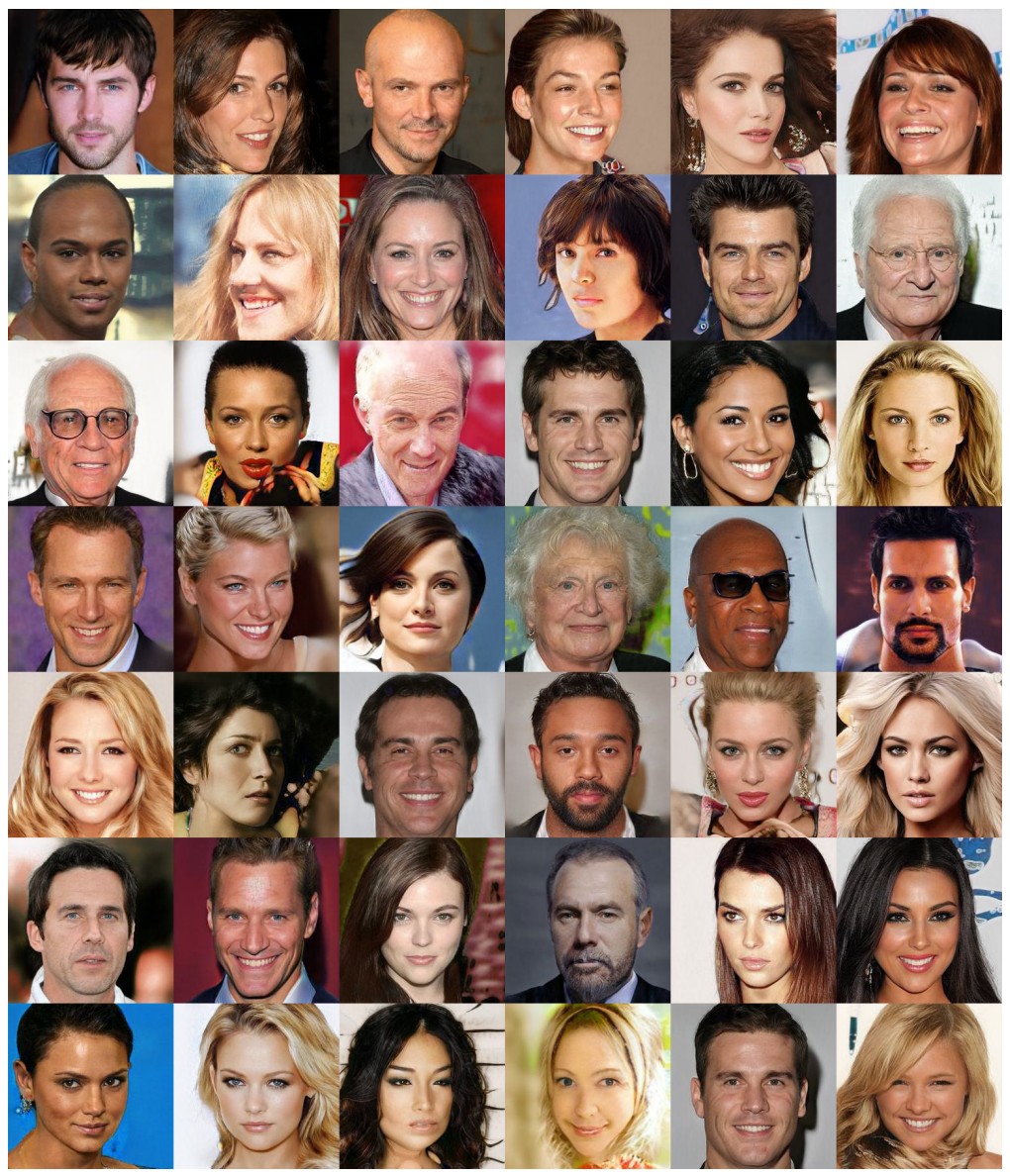

Figure 15: Generated samples from UOTM trained on CelebA-HQ ($256 \times 256$).

