# OpenReview forum: "Generative Modeling through the Semi-dual Formulation of Unbalanced Optimal Transport"
_NeurIPS.cc/2023/Conference — NeurIPS 2023 poster_

### Official Review · Reviewer_Pz9z · 2023-07-02

**Soundness:** 3 good
**Presentation:** 3 good
**Contribution:** 3 good
**Rating:** 6
**Confidence:** 3

**Summary:**

The authors propose to use semi-dual formulation of unbalanced optimal transport for generative modelling. Via the experiments on image datasets, the authors illustrate the advantages of the proposed method, namely outlier robustness, stability and fast convergence, and extensively compare with other methods.


**Strengths:**

The authors have spent significant effort on the experiments, including extensive comparison with other competing methods, abalation study.
The literature review is also good. To this extent, I'm convinced that UOTM is indeed an efficient method for generative modelling under the presence of outliers.

**Weaknesses:**

However, this work is a direct implementation/application of [67] with no additional improvement or modification. The training procedure is also standard in GAN. For this reason, I would say the overall contribution is quite incremental, and this puts me on the borderline decision, where I'm fighting between the novelty/originality of the work and the effort that the authors spend on the experiments.

- While the authors compare UOTM with various methods, I don't see the comparison with [76]. Is there any reason that [76] is not compared or considered in the experiments?

- Given the fact that unbalancedness implies robustness, it's not new nor surprising to see UOTM outperforms OTM under the presence of noise. Probably it's worth discussing more about the comparison with other unbalanced-based methods (e.g. [8] or [17]), rather than balanced-based methods.

**Questions:**

- I'm not very clear about the order of magnitude of $\tau$ in the experiment. Is it $0.5, 1.2$, or $0.5 \times 10^{-3}, 1.2\times 10^{-3}$. The smaller it is, the more we are forced to respect the marginal constraint, i.e. approaching the balanced OT problem.

- What is the definition of Softplus?

- Regarding the theoritical justification of outlier robustness of UOT, the authors may consider to add reference [17]. I see that it is already cited, but not for that purpose.


**Limitations:**

The authors do not discuss the limitations, but do discuss the potential negative societal impact of their work.

---

> ### Author Rebuttal · Authors · 2023-08-08
>
> We sincerely appreciate the reviewer for acknowledging that "UOTM is indeed an efficient method for generative modeling under the presence of outliers". Also, we agree with the reviewer that, in terms of method, our model is a direct parametrization of the semi-dual form of UOT [67]. Nevertheless, we would like to emphasize the contribution of our work.
>
> - Our work is the first attempt to introduce the semi-dual form of UOT into the generative modeling task. **The semi-dual form enables 2 network parametrizations of UOTM, while the other UOT-based approaches [1, 2] require 3 networks (Line 188-196).** We believe this factor simplifies the training process, contributing to the achievement of competitive results.
>
> - As the reviewer said, we demonstrated UOTM is an efficient method for generative modeling under the presence of outliers (Sec 5.1).
>
> - Moreover, we showed UOTM can be a competitive model for clean datasets as well. The soft constraint on distribution matching of UOT induces a concern about its performance on clean datasets. **To address this concern, we provided an upper-bound of the divergences between the marginals in UOT (Theorem 3.3). We evaluated UOTM on the image benchmark datasets, such as CIFAR-10 and CelebA-HQ (Sec 5.2).**
>
> - Furthermore, we provided **explanations for the observed experimental results where UOT-based model (UOTM) outperforms OT-based model (OTM) in terms of distribution matching, despite the soft constraint of UOT**. These explanations are in two folds:
>
>     - First, we suggested a theoretical explanation based on Theorem from [19]. UOT objective minimizes not only the gradient error $\\| \nabla (v^{c}-v^{\star c}) \\|$, but also the function errors $\\| v^{c}-v^{\star c}\\|, \\| v-v^{\star}\\|$ (Line 148-157). In contrast, OT objective only minimizes the gradient error $\\| \nabla (v^{c}-v^{\star c})\\|$. We hypothesize this property enables the neural network training easier for UOTM, leading to better distribution matching than OTM.
>
>     - Second, we provided some intuitive explanations from the engineering point of view in Appendix C.1. Compared to OTM, UOTM enables adaptive updates for each sample, with weaker updates on well-performing data and stronger updates on poor-performing data (Line 649-650).
>
> $ $
>
> ------
>
> > **W1.** While the authors compare UOTM with various methods, I don't see the comparison with [76]. Is there any reason that [76] is not compared or considered in the experiments?
>
> **A1.** We conducted experiments of [76] on CIFAR-10. Despite extensive efforts in exploring various hyperparameters, [76] provided non-competitive results (with architecture "Small"), with FID scores around 80. Note that UOTM achieves a FID score of 12.86 on CIFAR-10 with the same Small backbone network.
>
> $ $
>
> ------
>
> > **W2.** Given the fact that unbalancedness implies robustness, it's not new nor surprising to see UOTM outperforms OTM under the presence of noise. Probably it's worth discussing more about the comparison with other unbalanced-based methods (e.g. [8] or [17]), rather than balanced-based methods.
>
> **A2.** We appreciate the thoughtful advice. In the Outlier Robustness experiment (Sec 5.1), **our goal was to assess the robustness of the UOT-based model (UOTM) by comparing it with its OT-based counterpart (OTM).** Note that both UOTM and OTM are derived from the semi-dual form of each problem. Moreover, we aimed to provide a **better understanding of how the UOT-based model works** compared to the OT-based model.
>
> Instead, to demonstrate UOTM's superior performance among UOT-based models, **we compared UOTM with another UOT-based model (Robust-OT [8]) on image benchmarks such as CIFAR-10 (Table 1)**.
>
> $ $
>
> ------
>
> > **Q1.**
> I'm not very clear about the order of magnitude of $\tau$ in the experiment. Is it $0.5, 1, 2$ or $0.5 \times 10^{-3}, 1 \times 10^{-3}, 2 \times 10^{-3}$. The smaller it is, the more we are forced to respect the marginal constraint, i.e. approaching the balanced OT problem.
>
> **A3.** We apologize for the confusion. In the ablation study on $\tau$, $\tau$ is chosen among $\\{0.1 \times 10^{-3}, 0.5 \times 10^{-3}, 1 \times 10^{-3}, 2 \times 10^{-3}, 5 \times 10^{-3}\\}$. For clarity, we revised the description of the experimental setup in the manuscript as follows:
>
> - In Fig 7, our model maintains a decent performance of FID($\leq 5$) for $\tau = \\{0.5, 1, 2\\} \times 10^{-3}$.
>
> $ $
>
> ------
>
> > **Q2.**
> What is the definition of Softplus?
>
> **A4.** Thank you for the comment. To ensure self-containedness, we added the definition of Softplus in the appendix. The definition of Softplus is as follows:
>
> $$
>    \operatorname{Softplus}(x) = \log ( 1+ \exp (x) )
> $$
>
> $ $
>
> ------
>
> > **Q3.**
> Regarding the theoretical justification of the outlier robustness of UOT, the authors may consider adding a reference [17]. I see that it is already cited, but not for that purpose.
>
> **A5.** Thank you for the valuable advice. We added a reference [17] to support the outlier robustness of UOT as follows:
>
> - Line 87-88: Second, UOT can address the sensitivity to outliers, which is a major limitation of OT [17].
> - Line 212: One of the main features of UOT is its robustness against outliers [17].
>
> $ $
>
> **References**
>
> [8] Yogesh Balaji, Rama Chellappa, and Soheil Feizi. Robust optimal transport with applications in generative modeling and domain adaptation. NeurIPS, 2020.
>
> [19] Thomas Gallouët, Roberta Ghezzi, and François-Xavier Vialard. Regularity theory and geometry of unbalanced optimal transport. arXiv preprint arXiv:2112.11056, 2021.
>
> [76] KD Yang and C Uhler. Scalable unbalanced optimal transport using generative adversarial networks. ICLR, 2019.

---

> > ### Author Response · Authors · 2023-08-18
> >
> > We thank the reviewer for the efforts in reviewing our paper. We would appreciate it if the reviewer let us know whether our response was helpful in addressing the reviewer's concerns. If there are additional concerns or questions, please let us know.

---

> > > ### Comment · Reviewer_Pz9z · 2023-08-18
> > > **Response to Authors**
> > >
> > > I thank the authors for the response.
> > >
> > > It took me some time to read other reviews and their corresponding discussions. Thanks to them, I believe the paper contains more technical details than I thought, rather than just a direct application of [67]. So, the theoritical contribution seems to be much more solid than before. As I already mentioned in my review, the effort on the experiments is also significant. As long as the authors can integrate the feedbacks and discussions into the revised version, I think the  overall contribution is solid.
> > >
> > > The authors have also adequately answered my concern and questions. For this reason, I'm happy to increase my score to 6 and the contribution score to 3.

---

> > > > ### Author Response · Authors · 2023-08-18
> > > > **Thank you**
> > > >
> > > > We sincerely appreciate the reviewer for dedicating time to review our paper.

---

### Official Review · Reviewer_6D69 · 2023-07-03

**Soundness:** 2 fair
**Presentation:** 3 good
**Contribution:** 2 fair
**Rating:** 5
**Confidence:** 4

**Summary:**

The paper derives a semi-dual formulation for the unbalanced optimal transport problem (using the known dual reformulation) and solves it with neural networks. The most notable contribution of the paper is that this formulation is applied to generative modeling and achieved nearly SOTA results on several standard datasets. It is also claimed that proposed method is more robust, stable and converges faster than existing OT methods.

Despite the inspiring practical performance, it seems to me that the central theoretical result of the paper (which justifies the entire method) is incorrect. Overall, the paper lacks mathematical rigor and most derivations are done rather sparingly, raising the question whether they are actually correct and making it hard to check them. Plus there are some concerns regarding the experimental evaluations which I think should be more explicitly discussed.


**Strengths:**

- Novel extension of dual (semi-dual) optimal transport methods to the unbalanced setup.
- Good image generation results, code is available.
- Faster/more stable convergence than prior OT methods; various studies present on the effects of different parameters (phi, psi, tau, etc.).


**Weaknesses:**

- Incorrect main theoretical results, lack of mathematical rigor in the presentation.
- Insufficient clarity, vague experimental comparisons, limited details of main baselines.
- One may argue that the algorithmical/theoretical part of the paper is incremental as c-transform based approaches are rather widespread nowadays.

I discuss some weaknesses in detail below.

*Wrong results.* Overall, the paper lacks mathematical rigor. This is a very big issue as it seems to me that informal statements of theorems (such as “under the suitable assumptions”, etc.) have led to incorrect theoretical results included in the paper. To be precise, I think the main theoretical result which forms the basis for the proposed algorithm is incorrect. I mean Theorem 3.3. One of the results claimed there is that $T_{v^*}$ is the optimal map between the rebalanced distributions $\tilde{\mu}$ and $\tilde{\nu}$.

First, I do not understand here why the maximizer $v^*$ of the dual form actually exists (there is “sup” in all the dual forms, not “max”). This is not a trivial fact and should be proved if used. Second, I do believe that it is not true that $T_v^*$ is necessarily an optimal map even when $v^*$ exists. Let me try to provide a counterexample. Let $\mu=\delta_0$, $\nu=\delta_1$ and let $\phi_1=\phi_2$ for symmetry and consider cost $c(x,y)=|x-y|$. Due to symmetry, it is clear that $\tilde{\mu}=m\cdot delta_0$ and $\tilde{\nu}=m\cdot delta_1$ for some positive mass $m>0$. Hence optimal $v$ should be the optimal potential for the balanced transport problem between $\delta_0$ and $\delta_1$ (multiplied by mass $m$), here I follow eq. (28). The cost here is $1-0=1$ ($\times m$), and I suppose that $v(x)=x$ is an optimal potential: $v^c=-v$ and $v^c(0)+v(1)=0+1=1=$cost ($\times m$). At the same time, consider $T_v(x)=x+10000$. Then $|x-T_v(x)|-T_v(x)=-x=-v(x)=v^c(x)$, i.e., it is a minimizer in eq. (7). Clearly, $T_v$ maps $0$ to $10000$, so it is not an optimal transport map. Thus, I think the main theoretical result provided in the paper is not correct or may require additional assumptions.

I tried to check the proof to see where there may be a mistake but encountered some difficulties. There the authors rather sparingly use the calculus of variations on the space of function without any formal definitions of what are these “derivatives”, whether they actually exist and skip some related questions. I believe this lack of details (and, to some extent, informalism) may lead to even more mistakes or gaps in their results; maybe this part should be checked by an expert in this particular type of mathematical apparatus. Even assuming this part is true, I am sufficiently sure that there is a mistake in lines (29)-(31). I do not understand where from $T_{v^*}$ appears there (this is not explained); explanations in line 568-569 seem to be wrong.

Overall, this mistake which I pointed out seems to be very crucial for the paper: it turns out to be simply incorrect that the proposed method learns an OT map between distributions $\tilde{\mu}$ and $\tilde{\nu}$. Hence, it remains questionable what it actually learns and are there any guarantees.

One additional minor comment here: the function T is introduced in section 3.2 but there is no evidence, explanation or motivation explaining what this function is and how it is related to OT map (the answer only appears in the next section and seems not correct). The clarity of the exposition could be improved.

Taking into account my comment about the incorrectness of the theoretical results, I also do not understand how Theorem 3.4 of [19] explains better stability or performance. To be precise, the theorem is entirely about the dual problem (c-transform-based, function $v$ only), while the authors solve the saddle point problem with $v$ and $T$. Obtaining the better value for the dual necessarily means a better dual variable. Yet for me it is not clear how this relates to the stability as the actual problem is min-max and involves two functions ($v$ and $T$, and we are finally interested in $T$, yes?). While I think there is some meaningful idea in the discussion in lines 148-157, it still seems a little bit speculative and should be made more convincing.

*Limited details of comparisons.* The section with the comparisons is rather dense but still leaves some essential details in Appendix or even skips them. First, it was not evident for me how the authors applied their method to generative modeling (which transport cost has been used between the noise and real data). This (rather important) information seems to appear only in Appendix B.1 and, as far as I understand, the authors used the strategy similar to OTM* paper by Rout et. al. This could have been explained earlier. Second, it remains unexplained how the numbers in Table 1 are obtained (whether the authors reproduce all these methods or just took the metrics from the respective papers). Third, comparison with OT methods is somewhat vague. For example, the authors write that they compared with the OTM paper by Fan et. al. I have checked this paper and did not even find any experiments in image generation. Hence, I wonder how this comparison is done. Due to this, I can not judge how fair the comparisons are. I suggest the authors provide more details in the main text about all this.

*Related work.* The semi-dual neural OT methods (a.k.a. c-transform-based) already existed for several years. The authors cite only rather recent papers in the field and do not mention the early papers where these approaches appeared, e.g., [1,2,3].

[1] Henry-Labordere, P. (2019). (Martingale) Optimal Transport And Anomaly Detection With Neural Networks: A Primal-dual Algorithm. arXiv preprint arXiv:1904.04546.

[2] Nhan Dam, Q. H., Le, T., Nguyen, T. D., Bui, H., & Phung, D. (2019). Three-player wasserstein gan via amortised duality. In Proc. of the 28th Int. Joint Conf. on Artificial Intelligence (IJCAI) (pp. 1-11).

[3] Korotin, A., Li, L., Genevay, A., Solomon, J. M., Filippov, A., & Burnaev, E. (2021). Do neural optimal transport solvers work? a continuous wasserstein-2 benchmark. Advances in Neural Information Processing Systems, 34, 14593-14605.

*Misprints.*
- Line 99. Should be $R^d$ instead of $R$ (twice)?
- Line 556. Should be $C(X)$, $C(Y)$ I guess?
- Eq. (26). It seems like one has to assume that $\Psi^*$ and $\Phi^*$ are differentiable everywhere?


**Questions:**

- What is the point of solving generative modeling with optimal transport maps?
- Why is there no testing of U-OTM in the image generation with fixed first marginal (noise)? Can such an experiment be conducted? It would be interesting if there is an effect of relaxing the first marginal in your method, or it suffices to relax the second one only (for simplicity).
- How did the authors perform comparison with OTM (see weaknesses section)?
- It is known that the quality of the learned generative model significantly depends on the hyperparameters used. From remark 3.1 I see that the method OTM/OTM* is a particular case of the proposed unbalanced method with $phi^\star(x)=psi^\star(x)=x$. How did the authors perform the comparison with this method? Did the authors use the same code (e.g., your own) with the same hyperparameters (except for phi, psi)?
- Is T_v in lines 116-117 a measurable function?
- How do the authors measure KL divergence by using the samples (table 3)?
- Could you please somehow (empirically) show that your method indeed learns the unbalanced OT map?
- In lines 183-196, there is a discussion of some robust OT methods. Why is there no comparison with them? As far as I understand, this seems to be relevant (not absolutely sure).
- In the proof of Theorem A.1 I do not understand why the strong duality holds (max dual=min primal)? Could you please detail this?


**Limitations:**

Yes

**Update:** *reject* -> *borderline accept*

---

> ### Author Rebuttal · Authors · 2023-08-08
>
> We appreciate the reviewer providing valuable advice. We think answering the reviewer's comments has significantly improved our work. We hope our replies to be helpful in addressing the reviewer's concerns.
>
> ### **1. Theoretical concerns**
>
> $ $
>
> ### **1.1 Concerns already discussed in the manuscript**
> ---
> > **W1.** The paper lacks mathematical rigor. There are informal statements of theorems such as “under the suitable assumptions”.
>
> **A1.** As mentioned in Line 51-52, **we presented the detailed assumptions of probabilistic space and measure in Appendix A (Line 523-525).** Specifically, $\mathcal{X}$ and $\mathcal{Y}$ are assumed to compact complete metric spaces which are subsets of $\mathbb{R}^{d}$, and $\mu,\nu$ be positive Radon measures of mass 1 on $\mathcal{X}$, $\mathcal{Y}$, respectively.
>
> $ $
>
> ---
> > **Limited details 1.** First, it was not evident to me how the authors applied their method to generative modeling (which transport cost has been used between the noise and real data).
>
> **A2.** As mentioned in Line 59, we adopt the **quadratic cost** $c(x,y)=\frac{\tau}{2}\\|x-y\\|_{2}^{2}$ throughout this paper. To enhance clarity of implementation, we included the **training algorithm of UOTM** in Line 125-126 of the manuscript. Moreover, we provided a detailed explanation of the connection between OTM* (Rout et al.) and our UOTM in Line 178-179.
>
> $ $
>
> ---
> > **M3.** Eq 26, $\Psi^*$ should be differential everywhere.
>
> **A3.** As mentioned in Line 106 and 524, we assumed $\Psi^*$ s are differentiable.
>
> $ $
>
> ### **1.2 Corrections to Theoretical Components**
>
> ---
> > **W2.** There is no discussion on the existence of optimal $v$ and $T$. If exists, is $T_{v^*}$ an optimal map between the rebalanced distributions? The main theoretical result provided in the paper is not correct or may require additional assumptions.
>
> **A4.** We sincerely appreciate the reviewer for providing valuable feedbacks. We believe that these feedbacks have been a considerable help in revising our manuscript to provide a more mathematically rigorous statement. We added the additional assumptions on measures $\mu, \nu$ and $\Psi^{*}$ to ensure the existence of solutions, and corrected Theorem 3.3 as follows (We removed Eq (29)-(31) accordingly):
>
> $ $
>
> > **Theorem 3.3** Suppose that $\mu$ and $\nu$ are probability densities defined on $\mathcal{X}$ and $\mathcal{Y}$. Given the assumptions in Appendix A, suppose that $\mu, \nu$ are absolutely continuous with respect to Lebesgue measure and $\Psi^{\*}$ is continuously differentiable. Assuming that the optimal potential $v^{\star}=\inf\_{v\in\mathcal{C}}J(v)$ exists, $v^\star$ is a solution of the following objective
> $$
> \tilde{J}(v)= \int_{\mathcal{X}}{-v^c(x)d\tilde{\mu}(x)}+\int_{\mathcal{Y}}{-v(y)d\tilde{\nu}(y)},
> $$
> > where $\tilde{\mu}(x)={\Psi_1^*}'(-{v^\star}^c(x))\mu(x)$ and $\tilde{\nu}(y)={\Psi_2^*}'(-{v^\star}(y))\nu(y)$. Note that the assumptions guarantee the existence of optimal transport map $T^{\star}$ between $\tilde{\mu}$ and $\tilde{\nu}$. Furthermore, $T^\star$ satisfies
> $$
>     T^{\star}\in\mathcal{T}:=\\{T_{v^\star}:\mathcal{X}\rightarrow\mathcal{Y}\mid T_{v^\star}(x)\in\arg\min_{y\in\mathcal{Y}}\left[c(x,y) - v^{\star}(y)\right]\textit{ for measurable }T_{v^\star}\\}.\hspace{20pt} (a)
> $$
> > In particular, $D_{\Psi_1}(\tilde{\mu}|\mu)+D_{\Psi_2}(\tilde{\nu}|\nu)\leq\tau\mathcal{W}_2^2(\mu,\nu)$ where $\mathcal{W}_2(\mu, \nu)$ is a Wasserstein-2 distance between $\mu$ and $\nu$.
>
> $ $
>
> We assume $\mu,\nu$ are absolutely continuous and $\Psi^*$ is continuously differentiable. **These additional assumptions ensure the existence of the optimal potential $v^\star$ and optimal transport map $T^\star$ for the OT problem between $\tilde{\mu}$ and $\tilde{\nu}$.** The continuity of ${\Psi^*}'$ gives that $\tilde{\mu},\tilde{\nu}$ are absolutely continuous with respect to Lebesgue measure and have a finite second moment (Note that $\mathcal{X},\mathcal{Y}$ are compact. Hence, ${\Psi^*}'$ is bounded on $\mathcal{X},\mathcal{Y}$, and $\mathcal{X},\mathcal{Y}$ are bounded). These properties of $\tilde{\mu},\tilde{\nu}$ gives the existence of $v^\star$ and $T^\star$ [79].
>
> Moreover, thanks to the reviewer, through a rigorous investigation, we realized that $T_{v^\star}$ might not serve as a transport map. Specifically, **the optimal transport map $T^\star$ between $\tilde{\mu}$ and $\tilde{\nu}$ is contained in the set of introduced parametrization $\mathcal{T}$ in Eq (a) ([71], Remark 5.13)**. However, we cannot ensure the uniqueness of the introduced parametrizations. The suggested counterexample $T_{v}(x)=x+10000$ is also a problem of this uniqueness. Both optimal transport $T^\star (x)=x+1$ and $T_v (x)$ are included in the parametrization set, i.e., $T_{v},T^{\star}\in\mathcal{T}$. **Nevertheless, note that the introduced assumption of absolute continuity excludes the suggested counterexample of $\mu=\delta_0,\nu=\delta_1$. Moreover, the assumed cost function $c(x,y)=\|x-y\|$ is different from our case.**
>
> $ $
>
> ---
> > **W3.** It remains questionable if the proposed method learns OT map between $\tilde{\mu}$ and $\tilde{\nu}$.
>
> **A5.** In this part, we would like to advocate the $c$-transform-based parametrization (Eq (a)). **To the best of our knowledge, in the $c$-transform-based literature, various works [8, 37, 56, 76] employed this same parametrization of the OT maps $T$.** However, as discussed in [3, 4] and A4, the above parametrization does not provide a formal guarantee of convergence towards $T$.
>
> Nevertheless, as reported in [56], the suggested parametrization converges to $T$ in practice. We agree with the reviewer that this part remains a gap in the literature, and further investigation is required.  However, our contribution is not limited to the theoretical parts. We kindly ask the reviewer to take into account the experimental contributions of our study as well.
>
> $ $
>
> ---
> **Other concerns will be addressed in additional Official Comments.**

---

> > ### Author Response · Authors · 2023-08-10
> > **Additional Response (1/2)**
> >
> > ### **1.3 Other Concerns**
> > ---
> > > **W4.** Taking into account the comment about the incorrectness of the theoretical results, how does Theorem 3.4 of [19] explain better stability or performance? The paper actually solves the min-max problem of $v$ and $T$.
> >
> > **A6.** **The main objective of this paper is to propose the UOT-based generative model and analyze its various perspective as a generative model through experiments. We referenced Theorem 3.4 of [19] to explain a better target distribution matching of UOTM compared to OTM.** Considering that UOTM relaxes the hard constraint of OTM, we thought the better distribution matching of UOTM is an interesting phenomenon. Therefore, in this work, we suggested some possible hypotheses (explanations) that may contribute to the higher performance of UOTM:
> >
> > - The high performance could potentially be attributed to the stability of the potential $v$, as implied by Theorem 3.4, which "may" facilitate favorable convergence for UOTM. We carefully "hypothesized" (rather than "asserted") that it would be a reason for better convergence properties. Accordingly, we validated that UOTM shows better and faster convergence (Tab 1, 2, 3, and Fig 5).
> > - The higher performance could potentially be attributed to additional stability of the potential $v$, as implied by Theorem 3.4. We hypothesized that this stability of potential $v$ "may" provide favorable convergence in the saddle point problem of UOTM by stabilizing one factor. Thus, we empirically validated that UOTM indeed shows better and faster convergence (Tab 1, 2, 3, and Fig 5).
> > - We posit that real-world datasets could contain outliers, leading to UOTM exhibiting a more robust convergence (Fig 2,3)
> > - In Appendix C, we also discussed that UOTM might have some engineering advantages.
> >
> > Although the suggested explanations are hypotheses rather than solid theoretical justifications, we believe that our work is meaningful in offering insights into UOT-based generative modeling. Moreover, we believe these insights have the potential to greatly benefit future researchers in the field. We kindly request the reviewer to contemplate our contribution from this perspective.
> >
> > $ $
> >
> > ### **2. Questions and Minor Corrections**
> >
> > $ $
> >
> > ### **2.1 Concerns already discussed in the manuscript**
> > ---
> > > **M1.** Line 99, should it be $R^d$ instead of $R$?
> >
> > **A7.** No. $R$ is correct. Note that the convex conjugate is taken for the entropy function $f:[0,\infty)\rightarrow [0,\infty]$, and the domain of the entropy function is a subset of $R$.
> >
> > $ $
> >
> > ---
> > > **Q2.** Why is there no testing of UOTM in the image generation with fixed first marginal?
> >
> > **A8.** No, we tested UOTM with fixed first marginal on CIFAR-10. The result is presented in Tab 3 under the label Fixed-$\mu$.
> >
> > $ $
> >
> > ---
> > > **Q6.** How do authors measure KL divergence by using the samples Table 3?
> >
> > **A9.** As cited in Line 245 and discussed in Line 626-628, we adopted the k-Nearest-Neighbor based estimation [9].
> >
> > $ $
> >
> > ---
> > > **Q8.** In lines 183-196, there is a discussion of some robust OT methods. Why is there no comparison between them?
> >
> > **A16.** No, **we made a comparison with Robust OT [8] on CIFAR-10.** The result is presented in Tab 1. We conducted experiments of [76] on CIFAR-10. Despite extensive efforts in exploring various hyperparameters, [76] provided non-competitive results, with FID scores around 80. Note that UOTM achieves a FID score of 12.86 on CIFAR-10 with the same Small backbone network. Other UOT-based models [8, 10, 17, 44, 53] are discrete UOT algorithms. These discrete algorithms find transport plans between existing samples. Hence, these algorithms are not appropriate for generative modeling.
> >
> > $ $

---

> > ### Author Response · Authors · 2023-08-10
> > **Additional Response (2/2)**
> >
> > ### **2.2 Other Concerns**
> > ---
> > > **W5/Q3/Q4.** How did the authors perform a comparison with OTM (see weaknesses section)? What parameter did it use? I wonder how this comparison is done. Due to this, I can not judge how fair the comparisons are.
> >
> > **A10.** We thank the reviewer for the detailed comment. **All the scores of other models on CIFAR-10 and CelebA-HQ (Tab 1,2) are taken from their original papers, except for Fan et al. [76]** As the reviewer commented, Fan et al. [76] did not report image generation results. Hence, we implemented the algorithm of Fan et al. on our own, using the Large backbone model of our work. (Note that the Large backbone model is the same as ScoreSDE and DDGAN.) The training hyperparameters of Fan et al. are finetuned based on the hyperparameters of UOTM. **To ensure fairness of comparison, we would like to emphasize that UOTM (small) employs the same architecture as Robust OT [8] and OTM [56], and UOTM outperforms both methods.** We added $\dagger$ to indicate the results conducted by ourselves in Tables, and included descriptions to captions as follows:
> > - $\dagger$ indicates the results conducted by ourselves.
> >
> > $ $
> >
> > ---
> > > **W6.** Missing related works related to $c$-transform.
> >
> > **A11.** Thank you for the careful comment.  We added the suggested related works into the manuscript as follows in Line 116:
> > - We introduce $T_v$ to approximate $v^{c}$ following previous works [16, 56, 80, 81, 82].
> >
> > $ $
> >
> > ---
> > > **M2**  Line 556. $\mathcal{C}(X)$, $\mathcal{C}(Y)$.
> >
> > **A12.** Thank you for the corrections. We would incorporate corrections into the manuscript.
> >
> > $ $
> >
> > ---
> > > **Q1** What is the point of solving generative modeling with optimal transport maps?
> >
> > **A13.** As discussed in Lines 14-31, OT theory has been widely exploited in the field of generative modeling. In the beginning, WGAN introduced OT-based Wasserstein distance as a loss function. Recently, several works proposed employing the optimal transport maps between source and target distributions as a generative model.
> >
> > A thorough understanding of the benefits of OT map-based generative model remains elusive. Nevertheless, we think that these models have some strengths over GAN, which shares a similar adversarial training. In particular, we discovered OT-based models offer additional advantages in terms of stable convergence and mitigation of mode collapse in GANs.
> > As discussed in Line 315-316 and Appendix C, **we believe the cost function encourages the generated images $T(x)$ to disperse by aligning each image to the corresponding noise $x$, thereby preventing the mode collapse problem.** We consider that investigating the precise benefits of OT maps would be promising future work.
> >
> > $ $
> >
> > ---
> > > **Q5** Is $T_v$ in lines 116-117 a measurable function?
> >
> > **A14.** Yes, there exists such measurable $T_v$ by Prop 7.33 in [83].
> > We revised our manuscript in line 117.
> > - Note that $T_{v}$ is measurable ([8], Prop 7.33).
> >
> > $ $
> >
> > ---
> > > **Q7** Could you please somehow empirically show that your method indeed learns the unbalanced OT map?
> >
> > **A15. To the best of our knowledge, there is no explicit solution for the UOT objective (even for a 1D case). Consequently, we evaluated the validity of the transport plan through (i) monotonicity of the transport plan (Fig 2 and 4) (ii) KL divergence (Tab 3), and (iii) FID score.** For instance, the monotonicity of the transport plan (Fig 4) with low KL divergence result (for low $\tau$, Tab 3) shows that our transport plan is nearly optimal. Note that $T^\star$ of OT problem should be monotone increasing in Fig 2\&4 by Theorem 2.9 of [84]. Furthermore, given that $\Psi^*(x)=e^x$ (in Figures 2\&4), ${\rm{supp}}(\mu)={\rm{supp}}(\tilde{\mu})$ and ${\rm{supp}}(\nu)={\rm{supp}}(\tilde{\nu})$ (Theorem 3.3). Thus, $T^\star$ of UOT problem should also be monotone.
> >
> > $ $
> >
> > ---
> > > **Q9** In the proof of Theorem A.1, I do not understand why the strong duality holds.
> >
> > **A17.** With assumptions in Appendix A (such as convexity, monotonicity, and differentiability of $\Psi^*$) and assuming the existence of $v^\star$, the dual form of UOT satisfies the assumption of the Fenchel-Rockafellar theorem. The theorem implies that the strong duality holds. We added additional explanations in the proof of Theorem A.1 as follows (After Eq 23):
> >
> > - In other words,
> >     $$
> >     \sup_{(u,v)\in \mathcal{C}(\mathcal{X})\times \mathcal{C}(\mathcal{Y})} \left[\int_{\mathcal{X}} -\Psi_1^*(-u(x)) d \mu(x) + \int_{\mathcal{Y}} -\Psi_2^* (-v(y)) d \nu(y) - \imath (u+v \leq c) \right],
> >     $$
> >     where $\imath$ is a convex indicator function. Note that we assume $\Psi_1^*$ and $\Psi_2^*$ is convex, non-decreasing, and differentiable. Since $c\geq 0$, by letting $u \equiv -1$ and $v \equiv -1$, we can easily see that all three terms in the above equation are finite values. Thus, we can apply Fenchel-Rockafellar's theorem, which implies that the strong duality holds.

---

> > ### Author Response · Authors · 2023-08-10
> > **Response Reference**
> >
> > **References**
> >
> > [56] Rout, Litu, Alexander Korotin, and Evgeny Burnaev. Generative modeling with optimal transport maps. ICLR, 2022.
> >
> > [71] Villani, Cédric. Optimal transport: old and new. Vol. 338. Berlin: springer, 2009.
> >
> > [72] Wang, Qing, Sanjeev R. Kulkarni, and Sergio Verdú. "Divergence estimation for multidimensional densities via $ k $-Nearest-Neighbor distances." IEEE Transactions on Information Theory 55.5 (2009): 2392-2405.
> >
> > [76] KD Yang and C Uhler. Scalable unbalanced optimal transport using generative adversarial networks. ICLR, 2019.
> >
> > [78] Vacher, Adrien, and François-Xavier Vialard. "Semi-Dual Unbalanced Quadratic Optimal Transport: fast statistical rates and convergent algorithm." ICML, 2023.
> >
> > [79] Villani, Cédric. Topics in optimal transportation. Vol. 58. American Mathematical Soc., 2021.
> >
> > [80] Henry-Labordere, Pierre. "(Martingale) Optimal Transport And Anomaly Detection With Neural Networks: A Primal-dual Algorithm." arXiv preprint arXiv:1904.04546 (2019).
> >
> > [81] Nhan Dam, Quan Hoang, et al. "Three-player wasserstein gan via amortised duality." IJCAI, 2019.
> >
> > [82] Korotin, Alexander, et al. "Do neural optimal transport solvers work? a continuous wasserstein-2 benchmark." NeurIPS, 2021.
> >
> > [83] Bertsekas, Dimitri, and Steven E. Shreve. Stochastic optimal control: the discrete-time case. Vol. 5. Athena Scientific, 1996.
> >
> > [84] Santambrogio, Filippo. "Optimal transport for applied mathematicians." Birkäuser, NY 55.58-63 (2015): 94.

---

> > > ### Comment · Reviewer_6D69 · 2023-08-11
> > > **Answers to the answers**
> > >
> > > Thanks for clarifying many various details as well as some theoretical things plus pointing to some aspects which I might have overlooked. I also still have a few questions:
> > > - How to prove the newly established theorem (A4)? In my initial review, I also pointed to some parts in the current proofs which seem to be rather informal. It would be great to have a comment about how this will be fixed.
> > > - Your reply A2 seems not to fully answer my question. I understand that you use the quadratic cost. I still do not understand what is x and what is y in this cost? Is my understanding correct that x an image-size gaussian noise tensor and y are images (as in ODE/SDE based OT methods such as [1,2])? Sorry, but I did not find any clarifications about it (may be I missed something).
> > >
> > > [1] Liu, X., Gong, C., & Liu, Q. (2022). Flow straight and fast: Learning to generate and transfer data with rectified flow. arXiv preprint arXiv:2209.03003.
> > >
> > > [2] Chen, T., Liu, G. H., & Theodorou, E. A. (2021). Likelihood training of schrodinger bridge using forward-backward sdes theory. arXiv preprint arXiv:2110.11291.

---

> > > > ### Author Response · Authors · 2023-08-12
> > > > **Response(1/2)**
> > > >
> > > > We appreciate the reviewer for reading our rebuttal and for asking further questions.
> > > >
> > > > ---
> > > > > **Q1.** How to (formally) prove the newly established theorem (A4)?
> > > >
> > > > **A.** Following the reviewer’s suggestion, we provide the revised version of the statement and proof of Theorem 3.3.
> > > >
> > > > **Theorem 3.3**
> > > > Suppose that $\mu$ and $\nu$ are probability densities defined on $\mathcal{X}$ and $\mathcal{Y}$.
> > > > Given the assumptions in Appendix A, suppose that $\mu,\nu$ are absolutely continuous with respect to Lebesgue measure and $\Psi^*$ is continuously differentiable. Assuming that the optimal potential $v^\star = \inf_{v\in\mathcal{C}}J(v)$ exists, $v^\star$ is a solution of the following objective $$\tilde{J}(v) =  \int_{\mathcal{X}}{-v^c(x)d\tilde{\mu}(x)} +\int_{\mathcal{Y}}{-v(y)d\tilde{\nu}(y)}$$
> > > > where $\tilde{\mu}(x)={\Psi_1^*}'(-{v^\star}^c(x))\mu(x)$ and $\tilde{\nu}(y)={\Psi_2^*}'(-{v^\star}(y))\nu(y)$. Note that the assumptions guarantee the existence of optimal transport map $T^\star$ between $\tilde{\mu}$ and $\tilde{\nu}$. Furthermore, $T^\star$ satisfies
> > > > $$T^\star (x)\in\arg\inf_{y\in\mathcal{Y}}\left[c(x,y)-v^\star(y)\right],$$ $\mu-a.e.$. In particular, $D_{\Psi_1}(\tilde{\mu}|\mu)+D_{\Psi_2}(\tilde{\nu}|\nu)\leq\tau\mathcal{W}_2^2(\mu,\nu)$ where $\mathcal{W}_2(\mu,\nu)$ is a Wasserstein-2 distance between $\mu$ and $\nu$.
> > > >
> > > > $ $
> > > >
> > > > **proof.**
> > > > For any $(f,g)\in\mathcal{C}(\mathcal{X})\times \mathcal{C}(\mathcal{Y})$, let $$I(f,g)=\int_{\mathcal{X}}-\Psi_1^* (-f(x))d\mu(x)+\int_{\mathcal{Y}}-\Psi_2^* (-g(y))d\nu(y).$$Then, for arbitrarily given potentials $(u,v)\in\mathcal{C}(\mathcal{X})\times\mathcal{C}(\mathcal{Y})$, and perturbation $(\eta,\zeta)\in\mathcal{C}(\mathcal{X})\times\mathcal{C}(\mathcal{Y})$ which satisfies $u(x)+\eta(x)+v(y)+\zeta(y)\leq c(x,y)$,$$\mathcal{L}(\epsilon):=\frac{d}{d\epsilon}I(u+\epsilon\eta,v+\epsilon\zeta)=\int_{\mathcal{X}}\eta{\Psi_1^*}'(-\left((u+\epsilon \eta)(x)\right))d\mu(x)+\int_{\mathcal{Y}}\zeta {\Psi_2^*}'(-\left((v+\epsilon\zeta)(y)\right))d\nu(y).$$Note that $\mathcal{L}(\epsilon)$ is a continuous function. Thus, the functional derivative of $I(u,v)$, i.e. $\mathcal{L}(0)$, is given by calculus of variation (Chapter 8, Evans [85])$$\int_{\mathcal{X}}\delta u(x){\Psi_1^*}'(-u(x))d\mu(x)+\int_{\mathcal{Y}}\delta v(y){\Psi_2^*}'(-v(y))d\nu(y),$$where $(\delta u,\delta v)\in\mathcal{C}(\mathcal{X})\times \mathcal{C}(\mathcal{Y})$ denote the variations of $(u,v)$, respectively. Note that such linearization must satisfy the inequality constraint $u(x)+\delta u(x)+v(y)+\delta v(y)\leq c(x,y)$ to give admissible variations. Then, the linearized dual form can be re-written as follows:$$\sup_{(u,v)\in\mathcal{C(\mathcal{X})}\times\mathcal{C(\mathcal{Y})}}\int_X \delta u(x)d\tilde{\mu}(x)+\int_{\mathcal{Y}}\delta v(y)d\tilde{\nu}(y),\qquad\qquad (a)$$where $\delta u(x)+\delta v(y)\leq c(x,y)-u(x)-v(y)$, $\tilde{\mu}(x)={\Psi_1^*}'(-u(x))\mu(x)$, and $\tilde{\nu}(y)={\Psi_2^*}'(-v(y))\nu(y)$. Thus, whenever $(u^\star,v^\star)$ is optimal, the linearized dual problem is non-positive due to the inequality constraint and tightness (i.e. $c(x,y)-u^\star (x)-v^\star (y)=0$ $\pi$-almost everywhere), which implies that Eq (a) achieves its optimum at $(u^\star,v^\star)$. Note that Eq (a) is a linearized dual problem of the following dual problem:$$\sup_{u(x)+v(y)\leq c(x,y)}\left[\int_{\mathcal{X}}u(x)d\tilde{\mu}(x)+\int_{\mathcal{Y}}v(y)d\tilde{\nu}(y)\right].$$Hence, the UOT problem can be reduced into standard OT formulation with marginals $\tilde{\mu}$ and $\tilde{\nu}$. Moreover, the continuity of ${\Psi^*}'$ gives that $\tilde{\mu},\tilde{\nu}$ are absolutely continuous with respect to Lebesgue measure and have a finite second moment (Note that $\mathcal{X},\mathcal{Y}$ are compact. Hence, ${\Psi^*}'$ is bounded on $\mathcal{X},\mathcal{Y}$, and $\mathcal{X},\mathcal{Y}$ are bounded). By Theorem 2.12 in Villani et al. [79], there exists a unique measurable OT map $T^\star$ which solves Monge problem between $\tilde{\mu}$ and $\tilde{\nu}$. Furthermore, as shown in Remark 5.13 in Villani et al. [71], $T^\star(x)\in{\rm{arginf}}_{y\in\mathcal{Y}}\left[c(x,y)-v(y)\right]$ for $\mu-a.e.$.
> > > >
> > > > Finally, if we constrain $\pi$ into $\Pi(\mu,\nu)$, then the UOT objective $C_{ub}$ becomes $C_{ub}=\tau \mathcal{W}^2 (\mu,\nu)$. Thus, for the optimal $\pi \in \mathcal{M}+$ where $\pi_0=\tilde{\mu}$ and $\pi_1=\tilde{\nu}$, it is trivial that$$\tau\mathcal{W}^2 (\mu,\nu)\geq\tau\mathcal{W}^2(\tilde{\mu},\tilde{\nu})+D_{\Psi_1}(\tilde{\mu}|\mu)+D_{\Psi_2}(\tilde{\nu}|\nu),$$which proves the last statement of the theorem.

---

> > > > > ### Author Response · Authors · 2023-08-12
> > > > > **Response(2/2)**
> > > > >
> > > > > ---
> > > > > > **Q2.** What is x and what is y in this cost? Is my understanding correct that x an image-size Gaussian noise tensor and y are images (as in ODE/SDE based OT methods)?
> > > > >
> > > > > **A.**
> > > > > Thank you for providing clarification regarding the question. Initially, we thought the question was about the definition of cost function. As the reviewer understood, for the cost function $c(x,y)$, $x$ and $y$ are an image-size Gaussian noise tensor and images, respectively (as same as ODE/SDE based OT methods). In the original manuscript, this information was described in Appendix B.1. Following the reviewer’s advice, we revised Line 205 in the main text as follows:
> > > > > - Unless otherwise stated, we set $\Psi:=\Psi_1=\Psi_2$ and $D_\Psi$ as a *KL divergence* in our UOTM model (Eq 9). The source distribution $\mu$ is a standard Gaussian distribution $\mathcal{N}(\mathbf{0}, \mathbf{I})$ with the same dimension as the target distribution $\nu$.
> > > > >
> > > > > **References**
> > > > >
> > > > > [56] Rout, Litu, Alexander Korotin, and Evgeny Burnaev. Generative modeling with optimal transport maps. ICLR, 2022.
> > > > >
> > > > > [71] Villani, Cédric. Optimal transport: old and new. Vol. 338. Berlin: springer, 2009.
> > > > >
> > > > > [78] Vacher, Adrien, and François-Xavier Vialard. "Semi-Dual Unbalanced Quadratic Optimal Transport: fast statistical rates and convergent algorithm." ICML, 2023.
> > > > >
> > > > > [79] Villani, Cédric. Topics in optimal transportation. Vol. 58. American Mathematical Soc., 2021.
> > > > >
> > > > > [85] Evans, Lawrence C. Partial differential equations. Vol. 19. American Mathematical Society, 2022.

---

> > > > > > ### Comment · Reviewer_6D69 · 2023-08-13
> > > > > > **Thanks for the answers**
> > > > > >
> > > > > > Okay, finally, it seems like I understood the proof. Still I strongly recommend authors adding an additional background section/paragraph regarding the first variations on the functional spaces.
> > > > > >
> > > > > > Also, one more question: I checked Appendix B.1 to which you pointed to. There it is written that "*For the image generation task in Sec 5.2, we followed the source distribution embedding of Rout et al. [56] for the small model, i.e., the Gaussian distribution μ with the size of 3 × 8 × 8 is bicubically upsampled to 3 × 32 × 32.*". It seems to contradict to the previous answer about the full-scale noise. Could you please comment?

---

> > > > > > > ### Author Response · Authors · 2023-08-13
> > > > > > > **Thank you for the response**
> > > > > > >
> > > > > > > We express our gratitude to the reviewer for the precise and thoughtful advice.
> > > > > > >
> > > > > > > ---
> > > > > > > >  **Q1.** Still I strongly recommend authors adding an additional background section/paragraph regarding the first variations on the functional spaces.
> > > > > > >
> > > > > > > **A1.** Thank you for the valuable suggestion. We will add a lemma in the proof that addresses the first variations within the functional spaces.
> > > > > > >
> > > > > > > $ $
> > > > > > >
> > > > > > > ---
> > > > > > > > **Q2.** I checked Appendix B.1 to which you pointed to. There it is written that "For the image generation task in Sec 5.2, we followed the source distribution embedding of Rout et al. [56] for the small model, i.e., the Gaussian distribution μ with the size of 3 × 8 × 8 is bicubically upsampled to 3 × 32 × 32.". It seems to contradict to the previous answer about the full-scale noise. Could you please comment?
> > > > > > >
> > > > > > > **A2.** We apologize for the confusion. In the case of the large model, we used full-scale noise (Line 580-581). For the small model, to ensure a fair comparison, we adopted the same approach employed by Rout et al. [56] (Line 594-595). In specific, Rout et al. [56] introduced the fixed bicubic upsampling $Q$(3x8x8 -> 3x32x32) and considered the OT problem between $Q_{\\#} \mu$ and $\nu$. Hence, in this case, the cost function is measured between unsampled noise and image.

---

> > > > > > > > ### Comment · Reviewer_6D69 · 2023-08-14
> > > > > > > > **Answer**
> > > > > > > >
> > > > > > > > Okay. In the end, I find this paper interesting, showing promising practical results and theoretical statements. The initially submitted version of the paper contained mistakes in main theoretical results on which some successive statements partially depended upon. All these issues have been fixed and clarified by the authors during the rebuttal.
> > > > > > > >
> > > > > > > > Now I rate the paper as "**borderline accept**". Not higher because I can not see the actual final revision with all the issues fixed and it might be beneficial for the paper to go through a fresh round of reviews. Not smaller because the results presented in the paper are inspiring and worth showing to community.

---

> > > > > > > > > ### Author Response · Authors · 2023-08-14
> > > > > > > > > **Thank you.**
> > > > > > > > >
> > > > > > > > > We sincerely appreciate the reviewer for reviewing our paper. The insightful feedback from the reviewer has been a significant help in improving the quality of our manuscript. Thank you again for your valuable attention and contributions.

---

### Official Review · Reviewer_TBHC · 2023-07-06

**Soundness:** 3 good
**Presentation:** 2 fair
**Contribution:** 3 good
**Rating:** 7
**Confidence:** 3

**Summary:**

In this paper, the authors propose a novel model (UOTM) that uses (as an objective function) a semi-dual form of the unbalanced optimal transport (UOT) problem. Since UOT relaxes the hard constraint of OT on distribution matching, they also provide the theoretical upper bound of divergences between marginals. The experiments conducted show that UOTM outperforms existing OT-based generative models (OTM) on CIFAR-10 and CelebA-HQ-256, and achieves comparable results to the other state-of-the-art baselines. Moreover, UOTM is more robust to outliers than other OTMs and provides stable and fast convergence.

**Strengths:**

1. The proposed UOTM model has a solid theoretical background and appears original and significant.

2. The experimental setup is appropriate, and the obtained results prove the superiority of the proposed solution concerning the state-of-the-art.

3. UOTM is probably the first OT-based generative model that achieves state-of-the-art performance on real-world large-scale image datasets.

**Weaknesses:**

1. The presentation needs some improvement.

**Questions:**

Minor comments:

The captions in Tab. 1, 2, 3, 4 and Fig. 5, 6, 7 could be more informative.

Some figures (such as Fig. 5) are not referenced in the text.

The order of tables/figures needs to be changed (e.g., in the text, Table 1 is referenced after Table 3).

l. 52: "satisfies" --> "satisfy".

l. 67: "couplings" --> "coupling".

Eq. (4): ',' --> '.'.

Eq. (5): "inf" --> "sup" (already corrected in the supplement).

l. 107: ".)" --> "):".

l. 115: "sem-dual" --> "semi-dual".

l. 298: '.' --> ':'.

**Limitations:**

Yes.

---

> ### Author Rebuttal · Authors · 2023-08-08
>
> We are deeply grateful to the reviewer for reading our paper and offering thoughtful feedback.
>
> $ $
>
> ---
>
> > **Minor comments**
>
> **A.** Thank you for the valuable advice regarding the presentation of our work. Following the advice, we would revise our manuscript as follows:
>
> - The order and captions of tables/figures are revised.
>     - Table 1: Target Distribution Matching Test. UOTM achieves a better approximation of target distribution $\nu$, i.e., $ T\_{\\#} \mu \approx \nu$.
>     - Table 2: Image Generation on CIFAR-10.
>     - Table 3: Image Generation on CelebA-HQ.
>     - Table 4: Ablation Study on Csiszar Divergence $D_{\Psi_{i}}(\cdot | \cdot)$.
>     - Fig 5: FID Scores during Training on CIFAR-10.
>     - Fig 6: Ablation Study on Regularizer Intensity $\lambda$.
>     - Fig 7: Ablation Study on $\tau$ in $c(x,y)=\frac{\tau}{2} \\|x-y\\|_2^{2}$.
>
>
> - We included a reference to Fig 5 in Section 5.3 (Line 283).
>     - In CIFAR-10, UOTM converges in 600 epochs, whereas OTM takes about 1000 epochs (Fig 5).
>
> $ $
>
> ---
>
> > **Typos**
>
> **A.** We appreciate the reviewer's thoughtful comment. We would incorporate the recommended corrections into the manuscript.

---

> > ### Comment · Reviewer_TBHC · 2023-08-15
> > **Thanks for the response**
> >
> > I'm satisfied with the authors' rebuttal. I keep my score unchanged.

---

> > > ### Author Response · Authors · 2023-08-16
> > > **Thank you**
> > >
> > > Thank you for the response. We thank the reviewer for the comments and suggestions. We are glad to hear that the reviewer is satisfied with our rebuttal.

---

### Official Review · Reviewer_9t3v · 2023-07-06

**Soundness:** 3 good
**Presentation:** 3 good
**Contribution:** 2 fair
**Rating:** 6
**Confidence:** 4

**Summary:**

Standard optimal transport (OT) problem aims at comparing two probability distributions by finding an optimal coupling that achieves a minimum geometrical cost. A major bottleneck of standard OT is the equality of total transported mass between the underlying distributions.  This restraints based-OT data analysis on real machine learning pipelines among many other domains. To alleviate this problem, this constraint is relaxed and gives unbalanced optimal transport (UOT). This paper investigates based-UOT frameworks on generative modeling through semi-dual form of UOT.

**Strengths:**

- The paper is well-written and the approach is mostly well-presented. Addressing generative modeling through the UOT toolbox is novel up to my best knowledge. Numerical experiments in this paper show competitive performance of UOTM (UOT+generative modeling). A noted advantage of UOTM compared to OTM is its robustness to outliers and faster convergence.
- Semi dual formulation of UOT is given with a theoretical guarantee of its stability compared to other OT objectives (see Theorem 3.4).
- The code is attached in the supplementary and results are reproducible.

**Weaknesses:**

### Weakness/Questions
- In the primal formulation of UOT (see Eq 4), I noticed that in the divergence terms $D_{¨\Psi_1}$ and  $D_{¨\Psi_2}$ there are not penalized by some positive tuning parameters as well considered in many UOT problem, I mean that one can expect the following formulation:
$C_{ub}(\mu, \nu) = ... + \lambda_1D_{¨\Psi_1} + \lambda_2D_{¨\Psi_2}$ where $\lambda_1, \lambda_2$ control how much mass variations are penalized.
- I guess the $\tau$ hyperparameter has great importance in the given formulation of UOT via the cost $c(x, y) = \frac \tau2 \||x - y\||_2^2$:
   - (i) In the experiments $\tau$ should be as small as possible to get a good performance. Is the robustness of UOT is inherited from penalization of the cost or from properties of the divergence penalizations?
   - (ii)  Let's consider the same cost function $c(x, y) = \frac \tau2 \||x - y\||_2^2$ with the dual formulation rather than the semi-dual. Can we expect that UOT with dual has an approximate performance like semi-dual UOT?
- In L227 the mapping $T^\star$ should be monotone increasing (as shown in Figure 2, b). Could you please explain this fact?


**Questions:**

### Minor typos/suggestions

- L51: X and Y are compact complete spaces
- L78: Csiszar divergences (add a reference)
- L115: sem-dual --> semi-dual
- L122: in Eq (9) $v$ should be indexed by $\phi$

---

> ### Author Rebuttal · Authors · 2023-08-08
>
> We appreciate the reviewer for spending time reading our manuscript carefully and providing thoughtful feedback. We hope our replies to be helpful in addressing the reviewer's concerns.
>
> $ $
>
> ----
> > **Q1.** Introducing tuning parameters $\lambda$ to the divergence terms, i.e., $\lambda_{1} D_{\Psi_{1}} +  \lambda_{2} D_{\Psi_{2}}$.
>
> **A.** From the primal form of UOT below, **adjusting parameters $\lambda$ of the divergence terms is equivalent to adjusting the parameter $\tau$ in the cost function $c(x, y) = \frac{\tau}{2} \\| x-y \\|_{2}^{2}$.** Specifically, multiplying a constant $k>0$ to $\tau$ is equivalent to dividing both $\lambda_{1}$ and $\lambda_{2}$ by $k$. Instead of introducing tuning parameters to the divergence terms $D_{\Psi_1}, D_{\Psi_2}$, we controlled $c(x, y) = \frac{\tau}{2} \\| x-y \\|\_{2}^{2}$ by the parameter $\tau$. **The ablation study on $\tau$ is presented in Fig 7**.
> $$
>     C\_{ub}(\mu, \nu) := \inf\_{\pi \in \mathcal{M}\_{+}(\mathcal{X}\times\mathcal{Y})} \left[ \int_{\mathcal{X}\times \mathcal{Y}} c(x,y) d\pi(x,y) + D_{\Psi_1}(\pi_0|\mu) + D_{\Psi_2}(\pi_1|\nu) \right].
> $$
>
> $ $
>
> ----
> > **Q2(1).** In the experiments, $\tau$ should be as small as possible to get a good performance. Is the robustness of UOT inherited from cost or divergence penalization?
>
> **A.** **We consider that the robustness of UOT is inherited from the properties of divergence penalizations $D_{\Psi_1}, D_{\Psi_2}$.**
> In Sec 5.1, we evaluated the outlier robustness of UOT by comparing it with OT-based Models (OTM).
> Here, note that **OTM has the cost penalization term only, not the divergence terms.**
> Hence, if the robustness were due to the cost penalization, OTM would also exhibit robustness.
> However, the experimental results (Fig 2,3) contradicted this assumption by showing that OTM did not exhibit outlier robustness.
>
> $ $
>
> ----
> > **Q2(2).**
> Let's consider the same cost function with the dual formulation rather than the semi-dual. Can we expect that UOT with dual has an approximate performance like semi-dual UOT?
>
> **A.** We appreciate the insightful comment. In our opinion, **the dual form of UOT would be a more challenging problem in terms of neural network training** (As a reminder, we included the equation below). If we adopt the dual form and parametrize two potentials $u, v$ by two neural networks, we would not possess a direct parametrization of transport map $T$ (by neural networks). Moreover, the training process would need to handle an inequality constraint $u(x)+v(y) \leq c(x,y)$, which appears to be challenging,
>
> $$
> C\_{ub}(\mu, \nu) = \sup\_{u(x)+v(y) \leq c(x,y)} \left[ \int\_{\mathcal{X}} -\Psi^{\*}\_{1} (-u(x)) d \mu (x) + \int\_{\mathcal{Y}} -\Psi^{\*}\_{2} (-v(y)) d \nu (y) \right].
> $$
>
> $ $
>
> ----
> > **Q3.**
> In L227 the mapping $T^{\star}$ should be monotone increasing (as shown in Figure 2, b). Could you please explain this fact?
>
> **A.** Thank you for the comment. This is because, **for the 1D distribution case, the optimal transport map $T^{\star}$ takes the explicit form** for the quadratic cost function $c(x,y) = \frac{\tau}{2} \\| x-y \\|\_{2}^{2}$. Intuitively, **$T^{\star}$ maps the $i$-th largest source sample to $i$-th largest target sample.** Formally speaking, if we denote the CDFs (cumulative distribution function) of $\mu$ and $\nu$ as $F_{\mu}$ and $F_{\nu}$, $T^{\star} = F_{\nu}^{-1} \circ F_{\mu}$ ([78], Theorem 2.9). Therefore, $T^{\star}$ should be monotone increasing. We agree with the reviewer that this statement requires further explanation. We would revise our manuscript accordingly.
>
>
> - Note that the optimal $T^{\star}$ should be a monotone-increasing function. For the 1D distributions $\mu$ and $\nu$, the optimal $T^{\star}$ is a map that transports the $k$-percentile source sample to $k$-percentile target sample ($0 \leq k \leq 1$) for $c(x,y) = \frac{\tau}{2} \\| x-y \\|_2^{2}$ ([78], Theorem 2.9).
>
> $ $
>
> ----
> > **Minor typos/ Suggestions**
>
> **A.** Thank you for the careful advice. We would correct the manuscript accordingly.
>
> **References**
>
> [1] Santambrogio, Filippo. "Optimal transport for applied mathematicians." Birkäuser, NY 55.58-63 (2015): 94.

---

> > ### Comment · Reviewer_9t3v · 2023-08-15
> > **Thank you for the response**
> >
> > I thank the authors for their efforts in the rebuttal.

---

> > > ### Author Response · Authors · 2023-08-16
> > > **Thank you**
> > >
> > > Thank you for the response. We appreciate the reviewer for reviewing our paper.

---

### Decision · Program_Chairs · 2023-09-21

**Decision:**

Accept (poster)

**Comment:**

This paper introduces a novel and interesting generalized formulation of the Optimal Transport problem that relies on a semi-dual form of the unbalanced OT problem. The paper was well received by the reviewers in general; they highlighted in particular the strength of the experimental results, the quality of writing, and the reproducibility of the results based on the provided codebase.

Reviewer 6D69 had some concerns about the main theoretical result of the paper, unearthed through a very thorough and detailed read of the paper. Through a rich back-and-forth with this reviewer, the authors acknowledged the mistakes in the result but were able to identify the cause of the error and address it. The reviewer who raised this issue was mostly satisfied with the result. Thanks to reviewer 6D69, this paper has undergone thorough scrutiny, and assuming the authors will update their submission to fix the error and include a discussion summarizing the thread with the reviewer, this paper will come in a much stronger form, suitable for publication in the conference.